# Towards Unpredictable Worlds: Continual In-Context Reinforcement Learning in Non-Stationary Environments

## Abstract

Traditional In-Context Reinforcement Learning (ICRL) demonstrates impressive rapid adaptation, but its reliance on static environments limits its applicability. In contrast, real-world scenarios are inherently non-stationary, with continuous and unpredictable changes that challenge an agent's ability to adapt. To bridge this gap, we formally define and systematically investigate Continual In-Context Reinforcement Learning in Non-Stationary Environments. Our central question is: what model architectures and training strategies enable an agent not only to rapidly master new dynamics in a continuously evolving environment, but also to efficiently discard or isolate outdated information, thereby achieving robust online adaptation? To ground our investigation, we construct a new benchmark suite featuring two complementary non-stationary domains—a symbolic reasoning task and a physics-based control task—each modified to exhibit unpredictable, intra-lifetime dynamic changes. On these benchmarks, we conduct extensive evaluations at both the model and training-strategy levels. At the model level, we compare state-of-the-art sequence model architectures. At the training strategy level, we systematically analyze the influence of stationary versus non-stationary training, dynamic change frequency, context length, and interaction scale. Our findings demonstrate the necessity of non-stationary training and reveal critical factors shaping continual adaptation. These results provide actionable insights and design principles for building agents capable of learning and adapting in truly open and dynamic worlds.

## 1 Introduction

Reinforcement Learning (RL) provides a powerful paradigm for solving sequential decision-making problems through trial-and-error, and has achieved significant progress across diverse domains (Sutton et al., 1998). Recently, with the advent of powerful sequence models such as the Transformer (Vaswani et al., 2017) and its variants, an emerging approach known as In-Context Reinforcement Learning (ICRL) has shown immense potential (Laskin et al., 2022; Lee et al., 2023; Grigsby et al., 2023; Moeini et al., 2025). ICRL enables pretrained sequence models to emulate RL-like adaptation during a forward pass (Lin et al., 2023), purely by processing contextual information, such as historical reward-observation-action sequences, without updating network parameters. This gradient-free, test-time adaptation not only improves computational efficiency but also offers a promising step toward general-purpose agents capable of rapid multi-task adaptation.

However, prior ICRL research (Laskin et al., 2022; Grigsby et al., 2023; Team et al., 2023) generally assumes that while agents must adapt to different task instances, the underlying dynamics of any given instance remain fixed throughout its lifetime. In other words, during ICRL's adaptation phase, the "rules" of the environment are static. This stands in stark contrast to real-world scenarios, which are often non-stationary and unpredictable. Examples include gradual drifts in physical parameters, such as robotic component wear, and abrupt shifts in task rules or objectives, such as updates to game mechanics (Hamadanian et al., 2023). Such non-stationarity poses a critical challenge: agents must not only learn new dynamics but also avoid inference from outdated context, a problem akin to catastrophic forgetting, but occuring within a single lifetime.

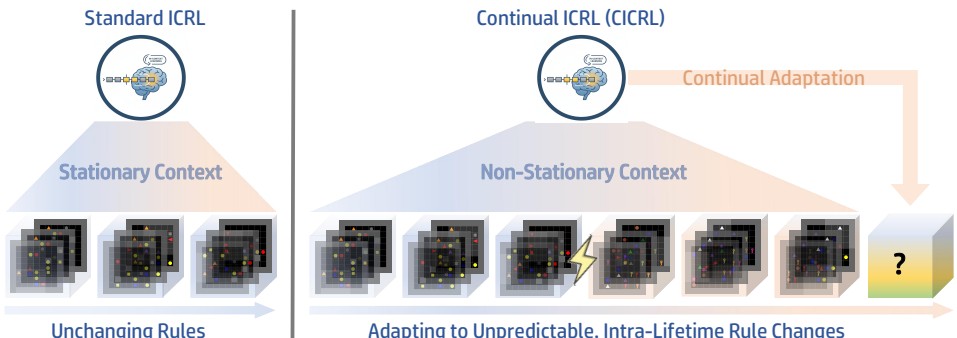

Figure 1: Comparison of Standard ICRL and Continual ICRL (CICRL). Standard ICRL adapts to fixed rules in a stationary environment, while CICRL requires an agent to continually adapt to unpredictable rule changes within a single lifetime.

Motivated by this gap, we explore the capabilities and limitations of ICRL in continuously non-stationary environments. We propose and investigate a central question: *when environmental dynamics shift within a single interaction sequence, can sequence models both rapidly adapt to new dynamics and effectively mitigate interference from obsolete context?* We refer to this extended setting as **Continual In-Context Reinforcement Learning (CICRL)**. As illustrated in Figure 1, unlike standard ICRL which adapts within a static environment with fixed rules, CICRL requires the agent to continuously adapt to unpredictable dynamic changes that occur within a single lifetime. CICRL requires agents to integrate the fast contextual adaptation strengths of ICRL with continual learning mechanisms that address intra-lifetime drift, thus bridging two previously separate lines of research.

To systematically investigate this problem, we first construct a new suite of non-stationary environment benchmarks. Specifically, we modify the rule-based XLand-Minigrid (Nikulin et al., 2024) grid-world suite and the physics-based Kinetix continuous control suite (Matthews et al., 2024b), introducing dynamic changes that occur within a single trajectory. These benchmarks compel agents to adapt online without relying on environment resets to clear its history. This intra-lifetime dynamic change provides a controllable platform for analyzing how sequence models manage context in the presence of evolving dynamics.

Using these benchmarks, we conduct comprehensive experimental evaluations and in-depth analyses of various advanced sequence model architectures on CICRL tasks. Alongside the widely-used Transformer (Vaswani et al., 2017), we evaluate emerging architectures that demonstrate strong performance in long-sequence modeling and computational efficiency, such as Mamba2 (Gu & Dao, 2023; Dao & Gu, 2024) and GatedDeltaNet (Yang et al., 2024b;a). Our analysis goes beyond final performance to deeply investigate their continual adaptation dynamics. We examine their speed of recovery from dynamic shifts, their resilience against interference from obsolete information, and how different training strategies shape these critical behaviors.

The main contributions of this paper include: 1) We formalize Continual In-Context Reinforcement Learning (CICRL), extending ICRL to address the critical challenge of intra-lifetime non-stationarity; 2) We introduce two novel non-stationary benchmarks for evaluating an agent's ability to adapt to and forget dynamic changes within a single lifetime; 3) We provide a systematic evaluation of modern sequence model architectures on CICRL tasks, revealing their strengths and limitations under non-stationary dynamics; and 4) We offer key insights into the continual learning dynamics of sequence models, highlighting open challenges and future directions for building more robust CICRL agents.

## 2 RELATED WORK

**In-Context Reinforcement Learning.** The concept of In-Context Reinforcement Learning (ICRL) is inspired by in-context learning in large language models (Brown et al., 2020; Li et al., 2023; Ruoss et al., 2024). Its core objective is to enable pretrained agents to rapidly adapt to new tasks by leveraging their interaction history, without requiring gradient updates to their network parameters (Laskin et al., 2022; Lee et al., 2023; Lu et al., 2023; Team et al., 2023). Early works such as

RL$^2$ (Duan et al., 2016) employ Recurrent Neural Networks (RNNs) as memory modules, training agents with standard reinforcement learning to implicitly learn a "fast learning algorithm." More recently, Transformer-based ICRL methods have demonstrated stronger adaptability and generalization potential. For instance, Algorithm Distillation (AD) (Laskin et al., 2022) enables Transformers to imitate the policy improvement process of a source RL algorithm by distilling its learning histories. Decision-Pretrained Transformer (DPT) (Lee et al., 2023) achieves a decision-making mechanism akin to posterior sampling by predicting optimal actions. AMAGO (Grigsby et al., 2023; 2024) successfully applies long-sequence Transformers to end-to-end RL by redesigning the off-policy actor-critic update, showing excellent performance in meta-learning and long-term memory domains. Although these studies represent significant progress in the ICRL field, they primarily focus on how agents adapt to different static instances within a broad task distribution. Less attention has been given to the continual adaptation capabilities of ICRL agents when the environment's dynamics change continuously within an agent's single lifetime (intra-lifetime).

**Reinforcement Learning Environments and Non-Stationarity.** The design of Reinforcement Learning (RL) environments is crucial for algorithm evaluation and development. Research progress has been driven by a progression from classic environments like Atari (Mnih et al., 2013) and MuJoCo (Todorov et al., 2012), to increasingly complex procedurally generated environments such as Procgen (Cobbe et al., 2020),NetHack (Küttler et al., 2020), XLand (Team et al., 2023), and Kinetix (Matthews et al., 2024b). These settings have continuously pushed research into agent generalization and adaptability. Furthermore, reconstructing environments like XLand-Minigrid (Nikulin et al., 2024) and Craftax (Matthews et al., 2024a) using frameworks such as JAX has significantly enhanced the efficiency of large-scale experiments. However, benchmarks specifically designed for systematically studying ICRL performance under non-stationarity, where the environment itself undergoes continuous unknown changes, remain scarce. Non-stationarity is a prevalent challenge in real-world settings, and while prior works have explored solutions (Hamadanian et al., 2023; Gospodinov et al., 2024), they often involve explicit online adaptation mechanisms. Similarly, methods like Domain Randomization (DR) (Tobin et al., 2017) improve robustness but typically yield static policies limited to average performance. In contrast, we investigate whether sequence models can address this challenge through inherent context-processing capabilities, enabling the agent to active infer and adapt to shifting dynamics online rather than relying on fixed weights.

**Sequence Models.** Sequence models, particularly the Transformer architecture based on self-attention mechanisms (Vaswani et al., 2017), have become powerful tools for processing sequential data. They have achieved revolutionary results in fields such as natural language processing (Brown et al., 2020), computer vision (Dosovitskiy et al., 2020), and reinforcement learning (Chen et al., 2021; Janner et al., 2021). With its parallel processing capabilities and effective capture of long-range dependencies, the Transformer provides a solid model foundation for ICRL. In recent years, a series of novel sequence architectures have been proposed to further enhance the efficiency and performance of long-sequence modeling. For instance, Mamba and its variants (Gu & Dao, 2023; Dao & Gu, 2024) introduce selective state space models, achieving linear-time complexity for long-sequence processing through hardware-friendly designs. DeltaNet and its gated versions (Yang et al., 2024b;a) enhance the model's associative memory and precise update capabilities by incorporating the delta rule. The development of these advanced sequence models offers a diverse range of architectural choices for constructing more powerful CICRL agents.

## 3 PROBLEM FORMULATION

In this section, we first present a formal formulation of standard In-Context Reinforcement Learning (ICRL), emphasizing its core assumption of static environments. Building upon this foundation, we introduce and define Continual In-Context Reinforcement Learning (CICRL), which explicitly captures the challenges faced by agents in non-stationary environments.

### 3.1 IN-CONTEXT REINFORCEMENT LEARNING IN A STATIC WORLD

The standard ICRL framework is built upon Partially Observable Markov Decision Processes (POMDPs). Each individual learning task corresponds to a POMDP instance, jointly defined by a state space $\mathcal{S}$, an action space $\mathcal{A}$, an observation space $\mathcal{O}$, a transition function $T$, a reward function $R$, an observation function $\Omega$, a discount factor $\gamma$, and a maximum horizon $H$.

In ICRL, the agent is exposed to a distribution of tasks $p(\mathcal{M})$. Its objective is to learn a single policy $\pi_\theta$ parameterized by $\theta$. This policy takes a continuous interaction history as context, denoted by $\mathcal{C}_t$, which contains the sequence of all observations, actions, and rewards from the initial time step up to the current time. Based on this context, the policy outputs the next action $a_t$. When encountering a new task instance sampled from the task distribution, the policy parameters $\theta$ remain fixed. The agent relies entirely on its growing context $\mathcal{C}_t$ to infer the characteristics of the current, previously unseen task and to rapidly adapt its behavior. A core assumption of this framework is intra-task stationarity: while an agent must be capable of adapting to different tasks, once a specific task instance $\mathcal{M}_i$ is selected, its underlying environmental dynamics, such as the transition function $T_i$ and reward function $R_i$, remain fixed throughout the agent's lifetime interacting with it. Consequently, adaptation in ICRL reduces to inferring unknown but constant environmental parameters from the observation history.

## 3.2 Continual ICRL in a Dynamic World

To investigate an agent's continual adaptation capabilities in a more realistic, ever-changing environments, we introduce the formal framework of Continual In-Context Reinforcement Learning (CICRL). The core feature of CICRL is intra-lifetime non-stationarity: environmental dynamics evolve continuously within a single interaction lifetime of the agent.

Unlike standard ICRL, we no longer assume a fixed distribution of task instances. Instead, the agent engages in a single, extended interaction within a continuously evolving environment. This evolution can be modeled as a sequence of POMDPs, $\{\mathcal{M}^{(k)}\}_{k=1}^K$, where during the $k$-th phase, the environment follows the dynamics of the POMDP instance $\mathcal{M}^{(k)}$, with transition and reward functions $T^{(k)}$ and $R^{(k)}$, respectively.

The fundamental difference from standard ICRL is that the transition from phase $k$ to $k + 1$ occurs within the agent's uninterrupted interaction, and the historical context $\mathcal{C}_t$ is never reset. Consequently, the context $\mathcal{C}_t$ may contain information from multiple, potentially conflicting environmental dynamics. As in ICRL, the agent's policy $\pi_\theta$ operates without gradient updates. However, its objective now is to maximize long-term cumulative return across this evolving sequence of POMDPs. This imposes additional demands: the policy must not only infer the current environmental dynamics $\mathcal{M}^{(k)}$ from the context, but also continuously track changes and mitigate interference from outdated information to achieve robust online adaptation.

## 4 The CICRL Benchmark Suite

To systematically investigate Continual In-Context Reinforcement Learning (CICRL), we construct a benchmark suite comprising two complementary environments, which cover both discrete, rule-based symbolic reasoning domains and continuous, physics-based dynamic control domains. We substantially modified both to exhibit intra-lifetime non-stationarity. This section details the intrinsic mechanics of these environments, our specific methods for introducing non-stationarity, and the algorithmic framework and evaluation protocols used in our experiments.

### 4.1 Rule-Based Non-Stationarity: The XLand-Minigrid Environment

Our first benchmark builds on XLand-Minigrid (Nikulin et al., 2024), a scalable, JAX-based grid-world environment capable of procedurally generating vast number of logically structured tasks.

**Core Mechanics and Agent Interface.** In XLand-Minigrid, each task is defined by a set of "Rules" and a "Goal." The rules act as the environment's "physical laws," defining how objects interact, e.g., "placing a blue pyramid ▲ next to a purple square ■ generates a yellow key 🔑." The agent must discover these hidden rules through trial and error and plan a sequence of actions to synthesize the goal object. Observations are symbolic partial grids, with each cell containing object type and color IDs. The action space is discrete, consisting of navigation and interaction commands.

**Introducing Intra-Lifetime Non-Stationarity.** In standard tasks, the rule-set remains fixed throughout the agent's lifetime. To introduce non-stationarity, we design an intra-lifetime rule evolution protocol, under this protocol, the agent interacts in a single, long session, and every $N_x$

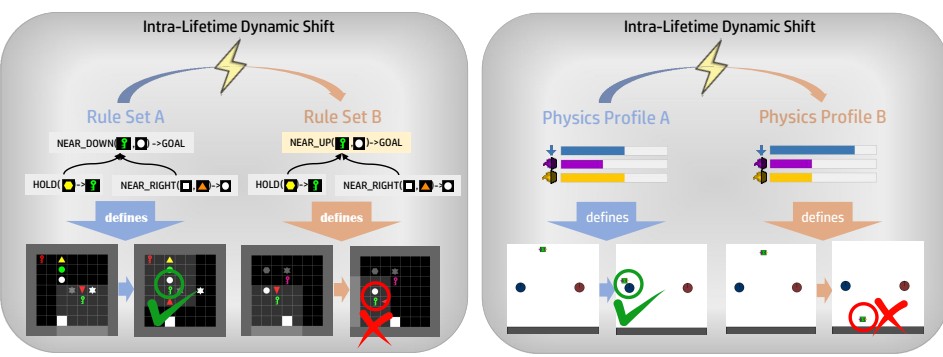

Figure 2: Examples of non-stationary dynamics in the CICRL benchmark suite. Left: In XLand-Minigrid, the environment's symbolic rule set changes intra-lifetime (e.g., the goal condition shifts from NEAR_DOWN to NEAR_UP), causing a previously successful policy to fail. Right: In Kinetix, underlying physical parametersthe underlying physical parameters (e.g., gravity or thruster power) are altered, which also invalidates the agent's learned behavior.

episodes, the environment's rule-set changes. As shown in Figure 2 (left), this evolution includes replacing key rules (e.g., the goal condition changing from 'NEAR_DOWN' to 'NEAR_UP'), as well as adding or removing "Distractor Rules." Agents must therefore continually infer which "physical laws" are currently active while disregarding outdated context.

## 4.2 PHYSICS-BASED NON-STATIONARITY: THE KINETIX ENVIRONMENT

To validate the generalization of CICRL to continuous control domains, our second benchmark builds on Kinetix (Matthews et al., 2024b), a JAX-based 2D physics simulation environment.

**Core Mechanics and Agent Interface.** A Kinetix scene is procedurally composed of fundamental physical components like rigid bodies, joints ▢, motors ⬣ and thrusters ◀. The agent controls actuated components (e.g., managing motor torques) to achieve goals such as colliding a green target object collide with a blue goal object. In this environment, agents observe raw visual pixels of the scene and output continuous control signals to drive the motors and thrusters.

**Introducing Intra-Lifetime Non-Stationarity.** We introduce non-stationarity into the environment on two levels: **structural layout** and **physical dynamics**. First, the layout of objects in the environment changes periodically. Second, we modify the underlying physics engine to enable dynamic changes in the environment's fundamental physical constants during an agent's lifetime. Parameters are randomized via a sampling mechanism at two levels: (i) **global dynamics** that affect the entire scene, such as gravity magnitude, thruster power, motor power, and base friction; and (ii) independent **object-specific properties** for each object, such as density, specific friction coefficients, and coefficients of restitution (elasticity). Every $N_k$ episodes, the system samples a new set of parameters, effectively placing the agent into a novel "physical reality". As illustrated in Figure 2 (right), every $N_k$ episodes, the system samples a new set of parameters (e.g., switching from "Physics Profile A" to "Physics Profile B"), effectively placing the agent into a novel "physical reality."

**Training and Test Ruleset.** A key feature of Kinetix is the distinction between its training and test sets. The training data is generated entirely through procedural synthesis. The test set, however, is based on a series of human-designed levels that test specific physical reasoning and control skills. To align this test set with our CICRL setting, we apply the same dynamic randomization of physical parameters to these human-designed levels. This evaluates whether agents can transfer dynamics inference abilities learned in procedural training to structured but physically unfamiliar tasks.

## 4.3 ALGORITHMIC FRAMEWORK AND EVALUATION PROTOCOL

To train and evaluate agents on these benchmarks, we employ a unified algorithmic framework and design a specialized evaluation protocol that analyzes their continual adaptation process across multiple dimensions.

**Training Framework.** Our training pipeline is based on the Proximal Policy Optimization (PPO) algorithm, integrated with a plug-and-play sequence model backbone. Non-stationarity is integrated directly into the training loop: after each dynamic phase, the environment manager samples a new duration $N \sim U(N_{\min}, N_{\max})$ for each parallel environment instance, where $N_{\min}$ and $N_{\max}$ are predefined episode counts. After this duration, the environment automatically switches its dynamic rules or physical parameters. This ensures that agents are continually exposed to dynamically paced changes during training.

**Evaluation Protocol.** Traditional average return metrics obscure the details of an agent's behavior during dynamic changes. To more finely characterize continual adaptation, we first define some notations. Assume the evaluation process runs in $D$ parallel environments for a total of $E$ episodes. We record all returns in a matrix $\mathbf{R} \in \mathbb{R}^{D \times E}$, where $R_{i,j}$ is the return of the $i$-th environment in the $j$-th episode. Let $M(i, j)$ denote the dynamics (e.g., a specific rule set or physics profile) in effect at episode $(i, j)$. Based on this, we design the following suite of evaluation metrics:

- **Zero-Shot Return (ZS-Return / ZS):** This metric measures the model's foundational capabilities without any valid historical context. We first compute the average zero-shot return $R_{ZS}(k)$ for each encountered dynamic $k$, which is the average performance when an agent first encounters dynamic $k$. The final ZS-Return is the macro-average of all these values. Let $\mathcal{S}$ be the set of all coordinates $(i, j)$ where a dynamic is encountered for the first time: $R_{ZS} = \frac{1}{|\mathcal{S}|} \sum_{(i,j) \in \mathcal{S}} R_{i,j}$
- **Sequence Return (Seq-Return / Seq):** This is the average of all elements in the return matrix $\mathbf{R}$, measuring the model's overall performance across the entire interaction sequence: $R_{Avg} = \frac{1}{D \cdot E} \sum_{i=1}^{D} \sum_{j=1}^{E} R_{i,j}$.
- **In-Context Delta ($\Delta_{\text{In-Context}}$):** This metric quantifies the average overall impact of contextual information. It is the mean difference between the return $R_{i,j}$ and the zero-shot return of its corresponding dynamic, $R_{ZS}(M(i, j))$: $\Delta_{\text{In-Context}} = \frac{1}{D \cdot E} \sum_{i=1}^{D} \sum_{j=1}^{E} (R_{i,j} - R_{ZS}(M(i,j)))$.
- **Switch Resilience ($\Delta_{\text{Switch}}$):** This metric specifically assesses the agent's resilience immediately after a dynamic shift. It measures the average difference between the return in the *first* episode after a switch and the ZS-Return of the new dynamic. Let $\mathcal{S}'$ be the set of all episode coordinates where a dynamic switch occurs (excluding initial episodes): $\Delta_{\text{Switch}} = \frac{1}{|\mathcal{S}'|} \sum_{(i,j) \in \mathcal{S}'} (R_{i,j} - R_{ZS}(M(i,j)))$. A non-negative $\Delta_{\text{Switch}}$ value indicates that the agent can effectively ignore outdated information and is robust to abrupt shifts.
- **Adaptation Gain ($\Delta_{\text{Adapt}}$):** This metric measures the agent's learning ability within a stable dynamic. It is the average gain in return for all *non-switch* episodes (i.e., the second episode and beyond within a dynamic), relative to the ZS-Return of that dynamic. Let $\mathcal{A}$ be the set of all non-switch episode coordinates: $\Delta_{\text{Adapt}} = \frac{1}{|\mathcal{A}|} \sum_{(i,j) \in \mathcal{A}} (R_{i,j} - R_{ZS}(M(i,j)))$.

Through this decoupled evaluation protocol, we can analyze the model's behavior from different dimensions, not only assessing its final performance but also revealing its learning, forgetting, and interference-resistance mechanisms when facing continual dynamic changes.

## 5 EXPERIMENTS

### 5.1 EXPERIMENTAL SETUP

**Environments and Training Protocols.** Our experiments are conducted in two distinct non-stationary environments: the rule-based **XLand-Minigrid** and the physics-based **Kinetix**. In XLand-Minigrid, the agent engages in a total of 10 billion (10B) interaction steps, while in the more computationally demanding Kinetix environment, the interaction budget is set to 1 billion (1B) steps. During both training and testing under non-stationary conditions, the frequency of environmental dynamic changes is set to occur after a random number of episodes, sampled uniformly from 1 to 10. This stochastic change frequency is designed to mimic real-world scenarios where the pace of environmental evolution is irregular. Unless otherwise stated in ablation studies, the interaction context length for all models during training is fixed at 25600 steps.

**Models.** We evaluate a diverse set of advanced sequence models, including Transformer, Mamba2, and GatedDeltaNet. To ensure a fair comparison, we configure each model with a similar number of layers and width. We employ a unified PPO training framework without introducing any architecture-specific modifications or enhancements. Additionally, we include a Domain Random-

ization (DR) baseline. This baseline is trained to learn a robust static policy across randomized environments, serving as a reference for standard generalization without explicit in-context adaptation. This controlled experimental setup allows us to directly assess the models' intrinsic continual adaptation abilities in the CICRL setting.

Table 1: Performance on **XLand-Minigrid**. Models trained under Static and Non-Static paradigms are evaluated in both static and non-static test environments.

| Model | ZS-Return | Static Env. Eval. | | Non-Static Env. Eval. | | | |
|---|---|---|---|---|---|---|---|
| | | Seq-Return | $\Delta_{\text{In-Context}}$ | Seq-Return | $\Delta_{\text{In-Context}}$ | $\Delta_{\text{Switch}}$ | $\Delta_{\text{Adapt}}$ |
| GatedDeltaNet (DR) | **0.201** | 0.164 | -0.035 | 0.104 | -0.097 | -0.143 | -0.090 |
| GatedDeltaNet (Static) | 0.084 | 0.188 | **0.100** | 0.081 | -0.003 | -0.012 | 0.000 |
| GatedDeltaNet (Non-Static) | 0.173 | **0.221** | 0.044 | 0.211 | **0.038** | 0.025 | 0.044 |
| Mamba2 (Static) | 0.089 | 0.166 | 0.078 | 0.093 | 0.007 | -0.012 | 0.009 |
| Mamba2 (Non-Static) | 0.190 | 0.216 | 0.027 | **0.213** | 0.027 | 0.009 | 0.028 |
| Transformer (Static) | 0.030 | 0.054 | -0.002 | 0.050 | -0.002 | 0.021 | 0.020 |
| Transformer (Non-Static) | 0.109 | 0.179 | 0.000 | 0.174 | 0.001 | **0.058** | **0.067** |

Table 2: Performance on **Kinetix**. The evaluation setup mirrors that of Table 1.

| Model | ZS-Return | Static Env. Eval. | | Non-Static Env. Eval. | | | |
|---|---|---|---|---|---|---|---|
| | | Seq-Return | $\Delta_{\text{In-Context}}$ | Seq-Return | $\Delta_{\text{In-Context}}$ | $\Delta_{\text{Switch}}$ | $\Delta_{\text{Adapt}}$ |
| GatedDeltaNet (DR) | -0.038 | -0.1204 | -0.093 | -0.122 | -0.084 | -0.058 | -0.085 |
| GatedDeltaNet (Static) | 0.023 | -0.020 | -0.019 | -0.023 | -0.002 | -0.038 | -0.051 |
| GatedDeltaNet (Non-Static) | 0.034 | **0.015** | 0.002 | -0.004 | 0.021 | -0.041 | -0.038 |
| Mamba2 (Static) | **0.035** | 0.005 | -0.005 | -0.003 | **0.024** | -0.042 | -0.038 |
| Mamba2 (Non-Static) | -0.065 | -0.080 | **0.004** | -0.008 | -0.070 | **0.057** | **0.056** |
| Transformer (Static) | -0.050 | -0.059 | -0.002 | **0.000** | -0.057 | 0.052 | 0.051 |
| Transformer (Non-Static) | 0.044 | **0.015** | 0.000 | -0.004 | **0.034** | -0.052 | -0.048 |

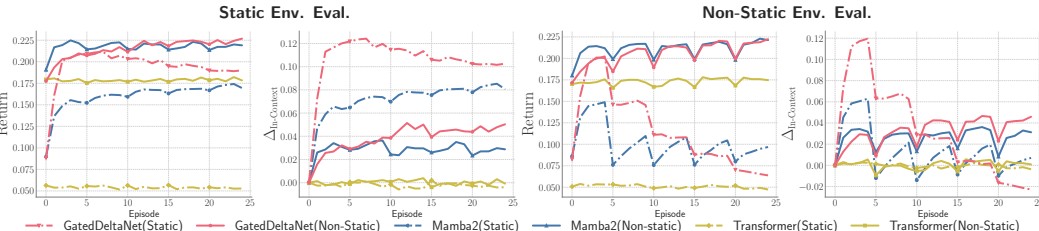

Figure 3: Results on XLand-Minigrid. In the non-stationary evaluation (right), models trained with non-stationarity (solid lines) successfully adapt to dynamic changes, while statically trained models (dashed lines) exhibit sustained low performance.

## 5.2 MAIN RESULTS: STATIC VS. NON-STATIONARY TRAINING

To systematically evaluate the necessity of non-stationary training, we compare models trained under **Static** and **Non-Stationary** paradigms. All models are evaluated in both static and non-stationary test environments, with detailed results presented in Table 1 and Table 2.

In the **XLand-Minigrid** environment, the advantages of non-stationary training are clearly validated, with the performance of GatedDeltaNet being particularly outstanding. As shown in Table 1, after non-stationary training, GatedDeltaNet's average return (Avg-Return) in the non-stationary test surged from 0.081 to 0.211, a significant increase that places it at a top-tier level among all models. The dynamic curves in Figure 3 visually illustrate this point: in the non-stationary evaluation (right plots), statically trained models (dashed lines) suffer a performance collapse after an environmental shift, whereas the non-stationarily trained GatedDeltaNet (solid red line) demonstrates strong recovery capabilities and consistently maintains a high level of performance. Although Mamba2 achieves a slightly higher average return, GatedDeltaNet shows highly competitive performance, proving that its architecture is well-suited for handling dynamic changes in rule-based, symbolic reasoning tasks.

Table 1 further highlights the distinction between static robustness and continual adaptation. The **GatedDeltaNet (DR)** baseline achieves the highest ZS-Return (0.201), outperforming the non-stationary model (0.173). This confirms that DR effectively learns a robust prior for the "average" environment. However, its lack of adaptation mechanisms leads to a collapse in long-term performance in the non-stationary test (Avg-Return drops to 0.104), with a significantly negative switch resilience ($\Delta_{\text{Switch}} = -0.143$). In contrast, our non-stationary training sacrifices slight initial zero-shot performance to enable rapid online adaptation, resulting in a much higher overall return (0.211) and positive adaptation metrics.

In the more challenging physics-based **Kinetix** environment, all models faced greater online adaptation pressure. Similar to XLand, the DR baseline struggles significantly here (Avg-Return $-0.122$), proving that static robustness is insufficient for complex, shifting physics. Yet GatedDeltaNet once again proved its robustness. As shown in Table 2, the difficulty of this task resulted in negative adaptation gains ($\Delta_{\text{Adapt}}$) for most models, indicating that context sometimes acted as a source of interference. However, under these demanding conditions, GatedDeltaNet achieved the highest average return (Avg-Return) alongside Transformer, showcasing its overall policy robustness in dealing with complex physical dynamics. Furthermore, it obtained a robust in-context gain ($\Delta_{\text{In-Context}}$), indicating its ability to effectively leverage contextual information. While Mamba2 exhibited unique performance on immediate post-switch adaptation metrics, GatedDeltaNet's ability to ensure strong, comprehensive long-term performance makes it an extremely reliable choice.

**Architectural Inductive Biases.** Comparing performance across domains reveals an intriguing pattern: SSM-based architectures like Mamba2 excel in the discrete, symbolic reasoning of XLand-Minigrid, likely due to their strength in modeling discrete state transitions. However, in the continuous control task of Kinetix, Transformer and GatedDeltaNet demonstrate superior robustness. This suggests that Transformer's direct access to historical tokens (via attention) and GatedDeltaNet's precise state rewriting mechanism (via the delta rule) may be better suited for handling the high-precision numerical adjustments required for physical adaptation, whereas standard SSMs may face challenges in retaining such fine-grained continuous information.

## 5.3 Ablation Studies: Key Factors Influencing Continual Adaptation

Building on the results above, we conduct a series of ablation studies in the non-stationary XLand-Minigrid environment, primarily using the GatedDeltaNet architecture. These experiments aim to identify the key hyperparameters and training factors that affect continual adaptation.

**Impact of Environmental Change Frequency.** We first examine how the frequency of environmental changes during training affects generalization to unseen non-stationary patterns. We train models with various frequencies: fast-random (1-5 episodes), slow-random (10-20 episodes), fixed (5 or 10 episodes), and our default balanced-random frequency (1-10 episodes). The results are shown in Figure 4. We evaluate all models in both a fast-changing (test frequency of 1) and a slow-changing (test frequency of 10) environment. The results highlight the importance of randomness. Models trained with a fixed frequency overfit to a specific rhythm and fail to generalize well to different change speeds. Among the randomized settings, overly rapid changes (1-5 episodes) may not allow the agent to fully learn from each dynamic, while overly slow changes (10-20 episodes) reduce the diversity of adaptation experiences. A randomized frequency of 1-10 episodes achieves the best trade-off, yielding the most robust performance across both fast and slow test scenarios.

**The Role of Context Length.** Context length is central to ICRL. To understand its impact in the CICRL setting, we train separate models using different context lengths: 6400, 12800, 25600, and 51200 steps. The results, shown in Figure 5, reveal that longer is not always better. Performance peaks at a context length of 25600 before degrading at 51200. From a training dynamics perspective, extremely long sequences can exacerbate the credit assignment problem for the PPO algorithm. The gradients from recent, more relevant transitions may be diluted, which could destabilize the learning process or necessitate adjustments to hyperparameters like the learning rate. The plots, which compare performance early (Episodes 1-50) and late (Episodes 151-200), confirm this trend is stable over long horizons. This finding suggests a "sweet spot" for context length and provides important practical guidance for designing CICRL agents.

**Scaling Effects of Training.** We further investigate how scaling along different dimensions influences model performance. Figure 6 presents the results for scaling interaction data and model size.

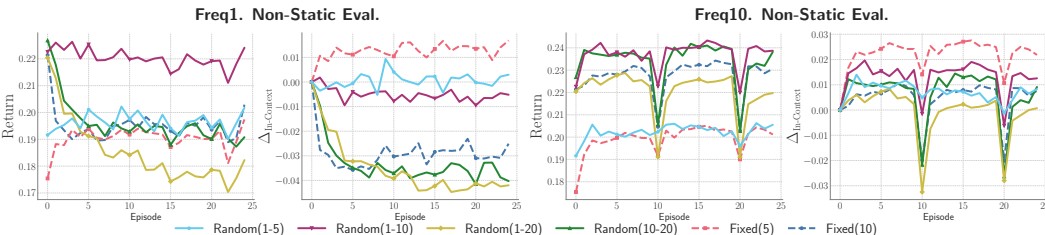

Figure 4: Impact of training with different dynamic change frequencies. Models are trained with fixed or randomized change frequencies and evaluated in environments with fast (Freq1, left) and slow (Freq10, right) changes. Training with a randomized frequency (Random 1-10) yields the most robust performance across both test settings.

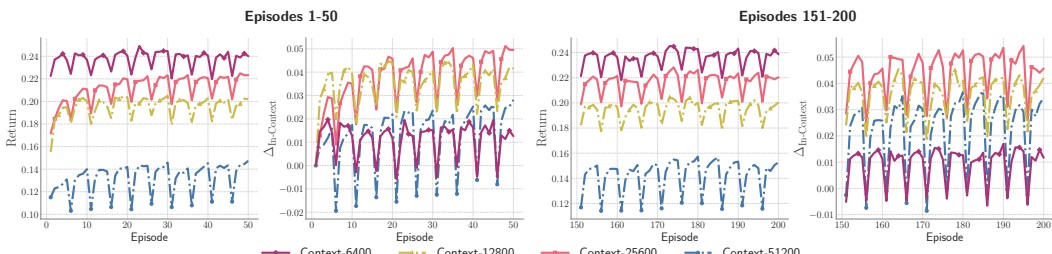

Figure 5: Effect of context length on continual adaptation. The plots show performance at the beginning (Episodes 1-50) and later (Episodes 151-200) of a long evaluation run.

**Interaction Data:** The left plots show a clear positive correlation between the amount of training data and performance. As the training budget increases from 1B to 20B steps, the model's average return and its ability to adapt after a switch steadily improve. **Model Size:** Similarly, the right plots show that increasing model depth from 3 to 6 layers leads to a significant and consistent improvement in continual adaptation. These results reveal a clear scaling trend, underscoring that CICRL benefits from scaling both data and model size.

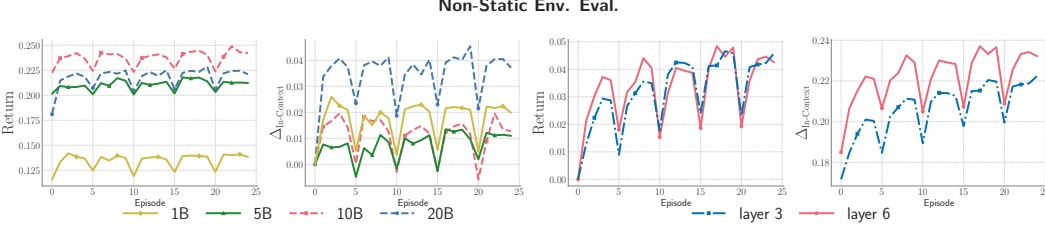

Figure 6: Scaling effects of training data and model size. **Left:** Performance improves as the total number of training interaction steps increases from 1B to 20B. **Right:** A deeper model (6 layers) consistently outperforms a shallower one (3 layers).

**Composite Change Patterns: "Stable" vs. "Chaotic" Eras.** In the Static-to-Fast setting (Figure 7, left), the environment remains stable for the first 25 episodes before entering a phase of rapid changes. The results show that while performance drops at the moment of the switch, the models quickly adapt to the new chaotic environment without collapsing. In the Fast-to-Static setting (Figure 7, right), the environment transitions from frequent changes to long-term stability. We observe that after the environment stabilizes at episode 25, the models' performance and in-context gains significantly improve and consolidate, demonstrating their ability to effectively exploit the new regularity.

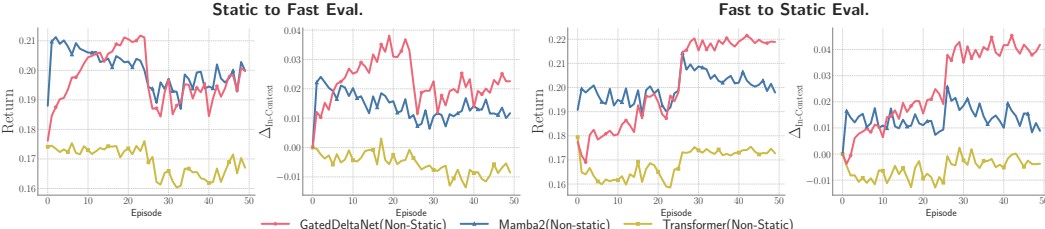

Figure 7: Robustness to composite dynamic changes. **Left (Static to Fast):** The environment is stable for the first 25 episodes before entering a phase of rapid changes. **Right (Fast to Static):** The environment starts with rapid changes and becomes stable after 25 episodes.

## 5.4 ROBUSTNESS TO CONTINUOUS AND STRUCTURED NON-STATIONARITY

Real-world dynamics often drift gradually rather than switch abruptly. To evaluate this, we introduced seven continuous and structured patterns. In **Kinetix**, physical constants evolve via *Linear*, *Step*, *Sinusoidal*, and *Sawtooth* functions. In **XLand-Minigrid**, we added *Depth Increase* (rising complexity), *Random Cosmetic* (symbol shuffling), and *AB Periodic* (looping rules).

We use $\Delta_{\text{In-Context}}$ to quantify the net impact of context, calculated as the difference between Sequence Return (**Seq**) and the Zero-Shot Return (**ZS**). Results in Table 3 (using GatedDeltaNet) reveal a critical flaw in static training: under continuous drift (Kinetix) or loops (XLand-AB), agents suffer *negative transfer* ($\Delta_{\text{In-Context}} < 0$), performing worse than zero-shot due to context interference. Conversely, Training on Non-Static environments enables effective

Table 3: Performance under Continuous/Structured Dynamics. $\Delta_{\text{In-Context}}$: Indicates the performance gain over ZS-R. Negative (red) implies interference; positive (blue) implies calibration.

| Domain | Pattern | Training on Static | | | Training on Non-Static | | |
|--------|---------|------|------|------------------------|------|------|------------------------|
| | | ZS | Seq | $\Delta_{\text{In-Context}}$ | ZS | Seq | $\Delta_{\text{In-Context}}$ |
| Kinetix | Linear | 0.108 | 0.096 | -0.012 | **0.113** | **0.114** | +0.001 |
| | Step | 0.087 | 0.074 | -0.013 | **0.116** | **0.121** | +0.006 |
| | Sinus | 0.103 | 0.096 | -0.007 | **0.115** | **0.127** | +0.012 |
| | Sawtooth | 0.100 | 0.099 | -0.001 | **0.104** | **0.112** | +0.008 |
| XLand | Depth | 0.218 | 0.257 | +0.039 | **0.303** | **0.315** | +0.012 |
| | Cosmetic | 0.148 | 0.162 | +0.014 | **0.232** | **0.240** | +0.008 |
| | AB Loop | 0.139 | 0.093 | -0.047 | **0.223** | **0.225** | +0.003 |

online calibration, tracking physical drifts and isolating conflicting rules (positive $\Delta_{\text{In-Context}}$), thus validating robustness to complex intra-lifetime dynamics. For detailed generation protocols and additional episode-wise analysis, please refer to Appendix C.2.

## 5.5 RELATIONSHIP WITH META-RL AND ONLINE RL

While CICRL shares the "learning to learn" objective with Meta-RL, it relies on *in-context* rather than *in-weight* adaptation. Gradient-based online methods (e.g., Online PPO), even when augmented with regularization techniques like EWC (Kirkpatrick et al., 2017), suffer from inherent update latency and instability under rapid intra-lifetime shifts. In contrast, CICRL freezes parameters to enable instantaneous adaptation via internal context states, avoiding the plasticity-stability dilemma of online updates. We validate this advantage in Appendix C.3, showing CICRL significantly outperforms both Online PPO and Online PPO+EWC.

## 6 CONCLUSION

In this paper, we formally introduced and systematically investigated Continual In-Context Reinforcement Learning (CICRL), extending the ICRL paradigm from static-task adaptation to the more realistic and challenging setting of intra-lifetime non-stationarity. Using our purpose-built non-stationary benchmark suite, we show that non-stationary training is crucial for overcoming contextual catastrophic forgetting and achieving robust continual adaptation. Our experiment further highlight that linear attention architectures, such as GatedDeltaNet, consistently outperform standard Transformers under dynamic conditions. Finally, our ablation studies reveal the critial role of factors such as context length and environmental change frequency, offering concrete insights for designing CICRL agents capable of adapting in complex, evolving environments.

ETHICS STATEMENT

This work focuses exclusively on synthetic and simulated environments—specifically symbolic reasoning and physics-based control domains—and does not involve human subjects, personal data, or sensitive content. Our research aims to advance the fundamental understanding of continual in-context reinforcement learning, without proposing direct real-world deployments that may carry immediate societal risks. Nevertheless, we acknowledge that reinforcement learning systems capable of continual adaptation could be applied in safety-critical domains. To mitigate potential risks, we emphasize that our contributions are intended as controlled benchmarks and analyses for academic research, not as ready-to-deploy systems. We are committed to releasing all code and benchmarks in an open and transparent manner to support reproducibility and responsible use. No conflicts of interest, external sponsorship, or ethical concerns beyond those discussed above are associated with this work.

REPRODUCIBILITY STATEMENT

We have taken several steps to ensure the reproducibility of our work: 1) For the benchmark suite, we provide detailed descriptions of the two non-stationary benchmark domains in the main paper and the supplementary material, including their environment dynamics, modification procedures, and change schedules. 2) For models and training, we specify the model architectures, parameterization strategies, training protocols, and hyperparameters in the paper. 3) We will release code and the benchmark environments to ensure reproducibility of our experiments. 4) All reported metrics and evaluation protocols are explicitly defined. Through these measures, we aim to make our work fully reproducible and to facilitate further research on continual in-context reinforcement learning.

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

APPENDIX

APPENDIX CONTENTS

## A  ENVIRONMENT AND MODEL DETAILS

This appendix provides an in-depth description of the environments, model architectures, and training parameters used in our experiments to ensure full reproducibility.

### A.1  ENVIRONMENT DETAILS

This section provides a detailed specification of the non-stationary environments designed for the Continual In-Context Reinforcement Learning (CICRL) benchmark.

#### A.1.1  NON-STATIONARY XLAND-MINIGRID

XLand-Minigrid is a grid-world domain designed to test symbolic reasoning and rule discovery capabilities.

**Observation and Action Space.** The agent interacts with the environment through a **discrete symbolic interface**.

- **Observation:** A partial $H \times W$ grid view, where each cell is represented by integer IDs encoding the object type (e.g., Triangle, Square) and color.

- **Action:** A discrete action space consisting of navigation (Move, Turn) and interaction (Pick Up, Put Down) commands.

Crucially, the observation **does not contain any explicit information** regarding the current rules or goal. The agent must infer the active "laws of physics" (e.g., which objects combine to form a key) solely from the interaction history in its context.

**Task Generation Logic.** We utilize a procedural generation pipeline to create diverse tasks based on a backward-chaining dependency tree:

1. **Goal Sampling:** A root goal is sampled (e.g., "AgentHold(Red Ball)" or "Near(Blue Square, Green Triangle)").
2. **Chain Expansion:** We recursively sample production rules that produce the required inputs for the current node (e.g., *Blue Key + Yellow Chest → Red Ball*).
3. **Pruning and Distractors:** To increase difficulty, branches are probabilistically pruned, and distractor objects (not involved in the solution) and distractor rules (valid rules leading to dead ends) are injected into the environment.

**Non-Stationary Protocol.** During a single lifetime, the environment transitions between different generated rulesets. Every $N \sim U(N_{min}, N_{max})$ episodes, the underlying logic changes. The agent must detect that previous successful strategies (e.g., combining Blue and Yellow) are no longer valid and adapt to the new rules.

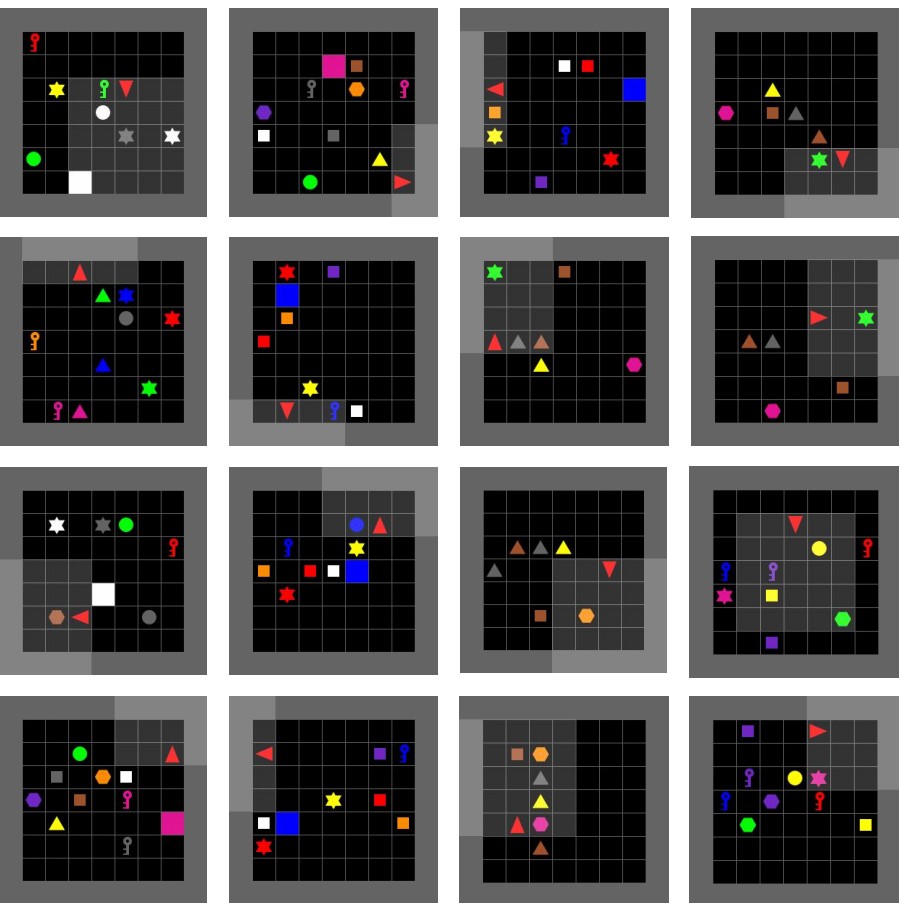

Figure 8: **XLand-Minigrid Environment.** The agent (red triangle) navigates a grid to manipulate objects. While the visual layout (walls, object positions) is randomized every episode, the underlying rules governing object interactions change only periodically, creating blocks of stationarity within a non-stationary lifetime.

### A.1.2 Non-Stationary Kinetix

Kinetix is a 2D physics-based simulation environment designed to evaluate motor control and physical reasoning.

**Observation and Action Space.** Unlike XLand, Kinetix operates in a continuous domain.

- **Observation:** The agent receives **pixel-based visual inputs** (images) representing the current state of the physics engine.
- **Action:** The agent outputs **continuous control signals** (torque and force) to drive motors and thrusters.

Similar to XLand, the agent is **never provided with the physical parameters** (e.g., gravity, friction coefficients) in the observation. It must infer the dynamics of the current "world" from how objects react to its actions.

**Environment Structure.** To evaluate generalization from unstructured exploration to structured tasks, we employ distinct configurations for training and testing:

- **Training (Unstructured):** As shown in Figure 9, training levels consist of procedurally generated, random graphs of polygons, joints, and thrusters. This prevents the agent from memorizing specific morphologies and forces it to learn fundamental physical interactions.

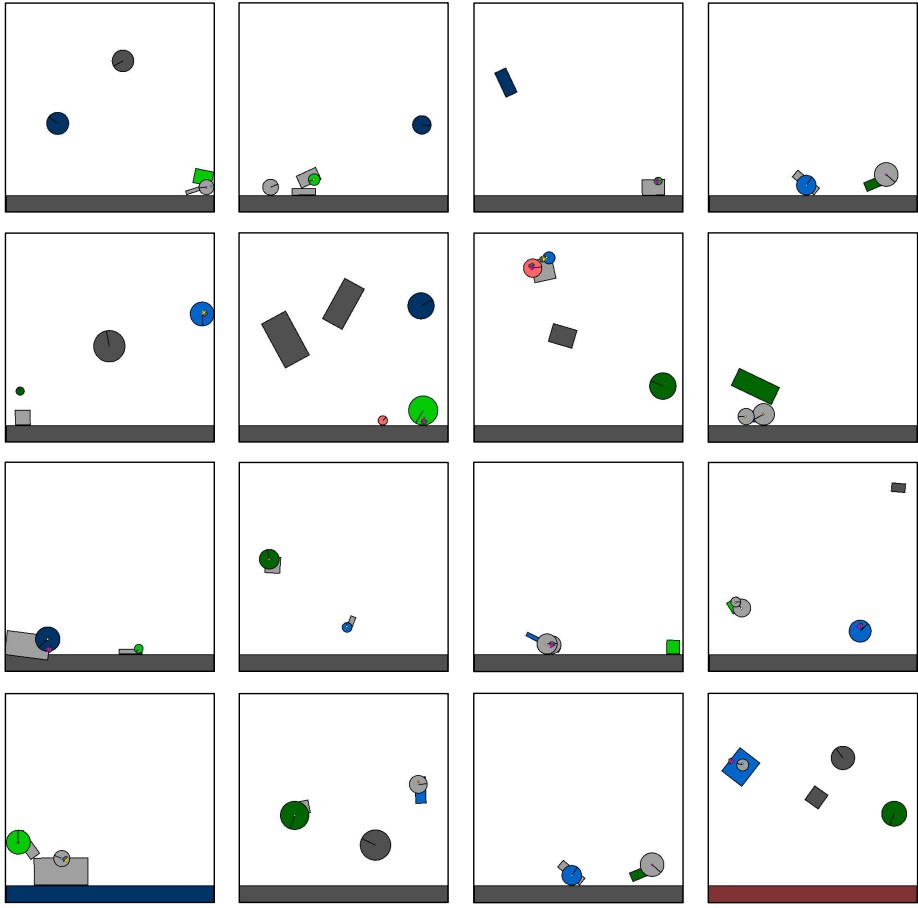

Figure 9: **Kinetix Training Environment.** An example of a procedurally generated, unstructured level used during training. The agent controls motors and thrusters to manipulate random morphologies.

- **Testing (Structured):** As shown in Figure 10, evaluation is performed on human-designed levels representing semantic tasks, such as bipedal walkers or robotic arms. This tests the transfer of learned physics adaptation to meaningful control tasks.

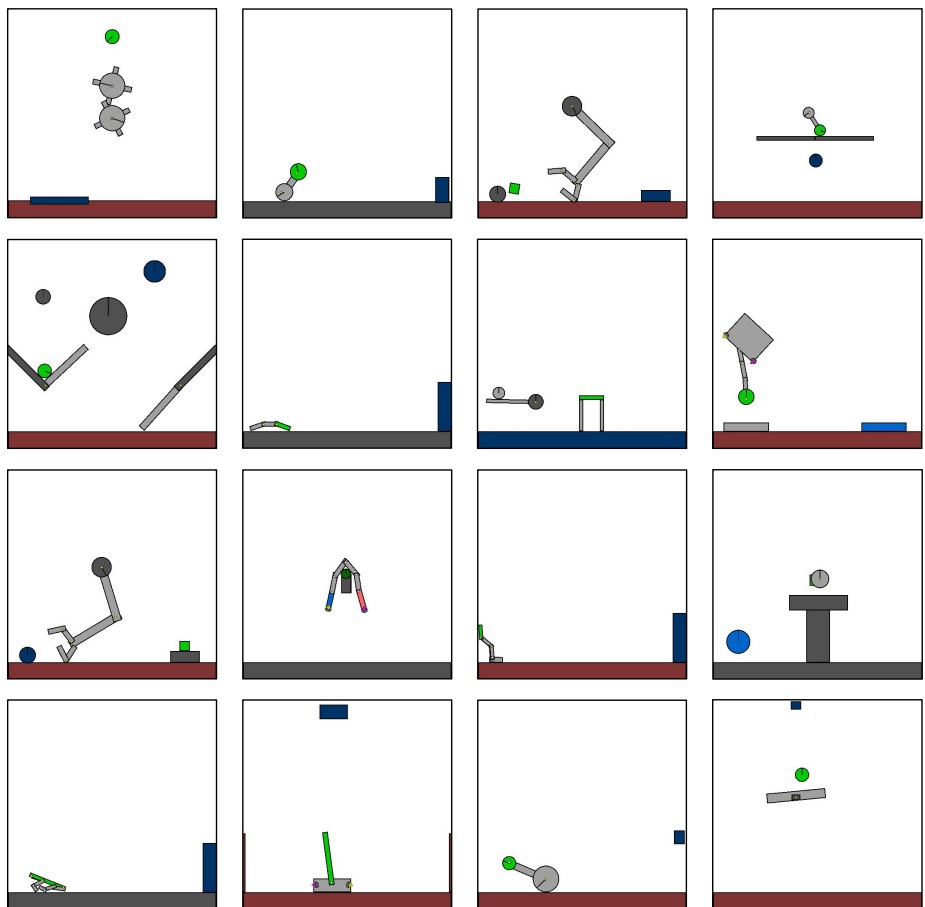

Figure 10: **Kinetix Test Environment.** An example of a structured evaluation level (e.g., a Walker task). Even in these structured tasks, the underlying physics (gravity, friction) change dynamically.

**Non-Stationary Dynamics Protocol.** To simulate an unpredictable physical reality, we apply a two-tiered randomization process every $N$ episodes. This modifies the internal parameters of the physics engine:

1. **Global Dynamics:** Parameters affecting the entire scene, such as the gravity vector, global air resistance, and global motor strength multipliers.
2. **Local Object Properties:** Physical coefficients including density, friction, and restitution (elasticity) are sampled independently for every object in the scene.

The specific sampling ranges for these parameters are detailed in Table 4.

## A.2 POLICY NETWORK ARCHITECTURE

To rigorously evaluate the Contextual In-Context RL (CICRL) capabilities across different architectures, we design a unified policy network framework. The model consists of a domain-specific time-step encoder (`TstepEncoder`), a sequence modeling backbone (Transformer, Mamba2, or Gated DeltaNet), and shared actor-critic heads. All models are standardized to a hidden dimension of $d_{model} = 256$ to maintain comparable capacity. In XLand-MiniGrid experiments, the total parameter count is approximately 2.6M, while in Kinetix experiments, it is approximately 4.3M. This

Table 4: Sampling distributions for Kinetix Physics Parameters. $U(a, b)$ denotes uniform sampling, $LogU$ denotes log-uniform sampling.

| Parameter Type | Parameter Name | Sampling Distribution |
|---|---|---|
| Global | Gravity Magnitude | $U(0.0, 20.0)$ |
| | Base Thruster Power | $LogU(\ln 0.2, \ln 5.0) \times 10.0$ |
| | Base Motor Power | $LogU(\ln 0.2, \ln 2.0) \times 900.0$ |
| | Joint Stiffness | $U(0.2, 1.0)$ |
| | Base Friction | $U(0.1, 2.0)$ |
| Per-Object (Local) | Density | $LogU(\ln 0.2, \ln 5.0)$ |
| | Friction Coefficient | $U(0.2, 1.5)$ |
| | Restitution (Elasticity) | $U(0.0, 0.6)$ |

difference arises from the larger feature extractors required for Kinetix's high-dimensional pixel observations.

### A.2.1 INPUT REPRESENTATIONS AND ENCODERS

At each time step $t$, the input to the model is a tuple $x_t = (o_t, a_{t-1}, r_{t-1})$, containing the current observation, the previous action, and the previous reward. The `TstepEncoder` processes these inputs into a dense vector $e_t \in \mathbb{R}^{d_{model}}$.

**XLand-MiniGrid Encoder (Symbolic & Discrete).** For the XLand-MiniGrid domain, observations are provided as $H \times W \times 2$ symbolic grids (object ID and color ID). We utilize an `EmbeddingEncoder` that maps discrete object and color indices to learned embeddings. These are concatenated and processed by a 3-layer Convolutional Neural Network (CNN) with $2 \times 2$ kernels and ReLU activations to extract spatial features. The previous discrete action $a_{t-1}$ is mapped via a learned lookup table (embedding layer). The scalar reward $r_{t-1}$ is unsqueezed and concatenated with the visual and action embeddings before a final linear projection.

**Kinetix Encoder (Pixel & Continuous).** For the Kinetix domain, observations are RGB pixel arrays. We employ a 2-layer CNN architecture: the first layer uses $8 \times 8$ kernels with stride 4, and the second uses $4 \times 4$ kernels with stride 2, similar to standard Atari encoders. Since Kinetix features a continuous action space, the previous action $a_{t-1}$ is encoded via a linear layer rather than an embedding table. The resulting flattened visual features, continuous action embeddings, and reward are concatenated to form the input to the sequence backbone.

**Actor-Critic Heads.** The output of the sequence backbone is fed into two separate heads. The **Critic** is a linear layer projecting to a scalar value. The **Actor** estimates a diagonal Gaussian distribution; it uses separate Multi-Layer Perceptrons (MLP) with Tanh activations to predict the mean $\mu$ and log-standard deviation $\log \sigma$.

### A.2.2 SEQUENCE MODELING BACKBONES

We compare three distinct backbones, each representing a different paradigm of sequence modeling. All backbones utilize a stack of $L = 3$ layers with a model dimension $D = 256$. To ensure efficient training and inference, all implementations of linear attention variants are based on the Flash Linear Attention (FLA) library[1].

**Transformer.** We employ a standard decoder-only Transformer with causal masking. The model is configured with 4 attention heads ($H = 4$, head dimension 64) and a feed-forward ratio of 4. The Transformer relies on the mechanism $\text{Attention}(Q, K, V)$, where "learning" manifests as retrieving relevant historical transitions via attention weights. This provides high-fidelity access to raw history but requires explicit storage of the Key-Value (KV) cache, scaling linearly with context length during inference.

---

[1]https://github.com/fla-org/flash-linear-attention

**Mamba2.**   We utilize the Mamba2 architecture, which bridges Linear Attention and State Space Models (SSMs) through the Structured State Space Duality (SSD) framework. Our configuration uses a state expansion factor of $N = 16$ and 16 heads (where the head dimension matches the state size). The core mechanism involves a data-dependent decay $\alpha_t \in (0, 1)$, governing the update rule $S_t = S_{t-1} \cdot \alpha_t + v_t k_t^\top$. In the context of CICRL, Mamba2 performs "learning" by updating the recurrent state $S_t$ and "forgetting" via the decay factor $\alpha_t$. When $\alpha_t \to 0$, the model exponentially suppresses historical information globally across the state. This allows for efficient $O(1)$ inference but may lead to information compression loss in long-horizon tasks requiring precise recall.

**Gated DeltaNet.**   We implement the Gated DeltaNet, a hybrid architecture combining the efficiency of RNNs with the expressivity of attention-like updates. The model is configured with 4 heads (head dimension 64) and utilizes a short convolution ($kernel = 4$) prior to the recurrent update. Distinct from the standard DeltaNet, Gated DeltaNet introduces a pivotal **data-dependent retention gate** $\alpha_t$, governing the update rule: $S_t = S_{t-1}(\alpha_t(I - \beta_t k_t k_t^\top)) + \beta_t v_t k_t^\top$. Here, $\alpha_t$ serves as the core forgetting component, enabling the model to dynamically decay the historical state based on current inputs. This works in tandem with the term $(I - \beta_t k_t k_t^\top)$, which provides "precise overwriting" capabilities. $\alpha_t$ facilitates rapid forgetting of outdated context (e.g., during task switches), whereas the delta rule allows for targeted updates of specific memory slots associated with key $k_t$. This dual mechanism offers a significant theoretical advantage in CICRL, balancing global adaptability with precise rule retention while maintaining $O(1)$ inference efficiency.

## A.3   TRAINING HYPERPARAMETERS

We employ the Proximal Policy Optimization (PPO) algorithm for all experiments. The training infrastructure is implemented in PyTorch, utilizing the FLA library for efficient sequence modeling. Optimization is performed using Adam with a fixed learning rate.

To handle the long-context requirements of CICRL, we utilize Truncated Backpropagation Through Time (TBPTT). The training process collects a large rollout buffer from parallel environments, which is then chunked into shorter sequences (defined by the BPTT horizon) for gradient updates.

Table 5 details the specific hyperparameters used for XLand-MiniGrid and Kinetix. While core PPO coefficients remain consistent across domains, simulation-specific parameters (e.g., number of environments, rollout length) are adjusted to accommodate the differing computational intensities and observation modalities of the two benchmarks.

Table 5: Detailed training hyperparameters for XLand-MiniGrid and Kinetix experiments.

| Hyperparameter | XLand-MiniGrid | Kinetix |
|---|---|---|
| *Optimization & PPO* | | |
| Total Timesteps | $1 \times 10^{10}$ | $1 \times 10^9$ |
| Learning Rate | $1 \times 10^{-3}$ | |
| Optimizer | Adam | |
| Discount Factor ($\gamma$) | 0.99 | |
| GAE Smoothing ($\lambda$) | 0.95 | |
| Clip Range ($\epsilon$) | 0.2 | |
| Entropy Coefficient | 0.01 | |
| Value Function Coefficient | 0.5 | |
| Max Gradient Norm | 0.5 | |
| Update Epochs | 1 | |
| Number of Minibatches | 64 | |
| *Rollout & Sequence Modeling* | | |
| Parallel Environments | 8192 | 1536 |
| Rollout Buffer Size (per env) | 25600 | 12800 |
| Truncated BPTT Horizon | 256 | 128 |

# B    THEORETICAL FRAMEWORK AND RELATED PARADIGMS

In this section, we provide a deeper theoretical analysis of Continual In-Context Reinforcement Learning (CICRL). We first clarify the mathematical relationship between CICRL and the standard Partially Observable Markov Decision Process (POMDP) framework, and then highlight the motivation for our specialized formulation. Finally, we distinguish the adaptation mechanisms of CICRL from traditional Meta-RL and Online Continual RL approaches.

## B.1    RELATIONSHIP WITH PARTIALLY OBSERVABLE MARKOV DECISION PROCESSES

**Mathematical Unification via Latent Variables.**    From a strictly probabilistic perspective, the sequence of evolving POMDPs $\{M^{(k)}\}_{k=1}^{K}$ defined in our problem formulation can be re-parameterized as a single, encompassing "Meta-POMDP" $\mathcal{M}$. By introducing an unobserved latent variable $z_t \in \mathcal{Z}$ to represent the current global environmental rules (e.g., the gravity vector or symbolic rule graph) at time $t$, the state space is augmented as $\mathcal{S}' = \mathcal{S} \times \mathcal{Z}$. The system dynamics evolve hierarchically:

$$
\begin{aligned}
z_t &\sim P(z_t \mid z_{t-1}) \\
s_t &\sim T(s_t \mid s_{t-1}, a_{t-1}, z_t) \\
o_t &\sim \Omega(o_t \mid s_t, z_t)
\end{aligned}
\tag{1}
$$

Here, $P(z_t \mid z_{t-1})$ defines the drift dynamics. Theoretically, the objective of maximizing long-term return in CICRL is mathematically equivalent to solving this augmented POMDP $\mathcal{M}$.

**Motivation for the Problem-Oriented Formulation.** Despite this mathematical equivalence, we explicitly formulate the problem as CICRL (a sequence of environments) rather than a standard POMDP. This modeling choice serves to precisely characterize the unique algorithmic challenges inherent to non-stationary settings, directly aligning with our benchmark design where blocks of stationarity are punctuated by shifts:

- **From State Estimation to System Identification (Structural Concept Drift).**    In standard POMDP applications, unobserved variables typically represent occluded *local states* (e.g., a hidden object). In contrast, variations in $z_t$ within CICRL represent fundamental shifts in the *Laws of Physics or Logic*, constituting **Concept Drift**. By framing this as a sequence of environments, we emphasize that the agent must perform online system identification and transfer learning across distinct "realities," rather than standard state estimation.
- **The Dual Nature of Context (Contextual Interference).** The most critical distinction lies in the utility of the interaction history $C_t$. In a static POMDP, history is strictly beneficial for reducing uncertainty. In CICRL, however, $C_t$ spans multiple phases of $z_t$ and inevitably contains *conflicting information* (e.g., an action rewarded in the past may be penalized now). Our formulation highlights that the context buffer is not merely a source of information but also a potential source of **interference**, shifting the core challenge from information aggregation to boundary detection and selective forgetting.

## B.2    DISTINCTIONS FROM META-RL AND ONLINE CONTINUAL RL

While CICRL shares the high-level objective of "learning to learn" with Meta-RL, it differs fundamentally in its information processing paradigm and adaptation mechanisms.

**Information Storage: In-Weight vs. In-Context.** Traditional Meta-RL (e.g., MAML) and Online Continual RL rely on **In-Weight Learning**. They attempt to encode new environmental dynamics by updating the neural network parameters $\theta$ via gradient descent. This creates a competition between new and old knowledge within the parameter space, often leading to catastrophic forgetting or requiring complex regularization (e.g., EWC) to mitigate instability.

In contrast, CICRL relies on **In-Context Learning**. We decouple the storage of meta-knowledge from local dynamics:

- **Frozen Weights ($\theta$):** Store universal, cross-task meta-knowledge (e.g., how to explore, how to deduce causal links). These parameters are *frozen* during deployment.
- **Fluid Context ($C_t$):** Stores the specific, short-term local dynamics of the current environment.

This decoupling allows CICRL agents to adapt to high-frequency changes (within milliseconds) without the instability or latency associated with online gradient updates.

**Mechanisms of Learning and Forgetting without Gradients.** A critical question in CICRL is how "learning" and "forgetting" occur without parameter updates. We formalize this through the lens of activation dynamics:

- **Activation-based Learning.** Although $\theta$ is fixed, the policy $\pi_\theta(a_t \mid C_t)$ is a dynamic function of the context. As new observations $(s, a, r)$ enter the sequence, the model's internal state (e.g., the KV cache in Transformers or the recurrent state $S_t$ in Linear Attention models) evolves recursively. This internal state shift drives the policy adaptation, a process theoretically shown to emulate gradient-based optimization steps within the forward pass. (Garg et al., 2022; Von Oswald et al., 2023; Wang et al.)
- **Mechanistic Forgetting.** To handle non-stationarity, the model must implement forgetting mechanisms without negative gradients:
  - **Attention Re-weighting (Transformers):** Through non-stationary training, the model learns to utilize the Softmax mechanism to dynamically down-weight attention scores assigned to tokens from previous, conflicting phases. This acts as a "soft" forgetting.
  - **Gated Erasure (Linear Attention/SSMs):** Architectures like GatedDeltaNet and Mamba2 implement "hard" forgetting via data-dependent decay gates. For instance, the update rule $S_t = \alpha_t S_{t-1} + (1 - \alpha_t \dots)v_t$ allows the model to generate a decay factor $\alpha_t \to 0$ upon detecting a reward shift. This effectively *flushes* the recurrent state $S_t$, removing outdated information and allowing the Delta Rule to *overwrite* the state with new dynamics.

Our experiments demonstrate that while standard sequence models possess these architectural capabilities, they only emerge as effective adaptation strategies when explicitly trained under the non-stationary protocols defined in our benchmark.

## C EXTENDED EXPERIMENTS

### C.1 COMPARISON WITH DOMAIN RANDOMIZATION

We compare the performance of Domain Randomization (DR), Static ICRL, and Non-Stationary CICRL. DR trains a robust policy on randomized environments without explicit context-based adaptation. Static ICRL uses long context but is trained on stationary tasks. Non-Stationary CICRL (Non-Static) is trained on dynamically evolving environments.

**Evaluation Setup.** To rigorously evaluate the adaptation capabilities, we conducted experiments in a non-stationary test environment where the underlying dynamics (e.g., symbolic rules in XLand or physical parameters in Kinetix) change abruptly every **5 episodes**. The agents are not explicitly notified of these boundaries and must infer the shifts solely from their interaction history.

**Macro-Performance.** Table 10 compares Zero-Shot (ZS) and Sequence (Seq) returns. In XLand-Minigrid, DR achieves the highest ZS return (0.201), confirming a strong prior, but its sequence performance degrades ($\Delta_{\text{In-Context}} = -0.098$). In Kinetix, however, DR struggles even at zero-shot ($-0.038$), likely due to overfitting the unstructured training distribution, which hinders generalization to structured test tasks. Meanwhile, Static ICRL exhibits negative transfer ($\Delta_{\text{In-Context}} < 0$) due to context interference. Non-Stationary training is the only approach yielding robust adaptation and stability across both domains.

Table 6: **Macro-Performance Comparison.** Zero-Shot (ZS) vs. Sequence (Seq) Return. $\Delta_{\text{Seq}}$ denotes the adaptation gain (Seq $-$ ZS).

| Method | XLand-Minigrid | | | Kinetix | | |
|---|---|---|---|---|---|---|
| | ZS-Return | Seq-Return | $\Delta_{\text{In-Context}}$ | ZS-Return | Seq-Return | $\Delta_{\text{In-Context}}$ |
| DR | **0.201** | 0.104 | -0.098 | -0.038 | -0.122 | -0.084 |
| Static | 0.084 | 0.117 | +0.033 | 0.023 | -0.004 | -0.027 |
| **Non-Static** | 0.172 | **0.206** | **+0.034** | **0.034** | **0.025** | **-0.009** |

**Micro-Dynamics.** Table 11 details the adaptation trajectory ($\Delta_t = R_t - R_{\mathrm{ZS}}$) across the 5 episodes of a specific dynamic phase.

- **XLand-Minigrid (Left):** DR consistently degrades over time (e.g., Rule 01: $-0.143 \rightarrow -0.085$), indicating a mismatch between the static policy and the specific rule. Static ICRL improves on simple rules but stagnates on complex ones (Rule 04). Non-Static training shows consistent positive learning curves.
- **Kinetix (Right):** DR and Static ICRL suffer from severe interference (negative $\Delta_t$) due to mismatches in physical parameters. Non-Static training effectively mitigates this interference, maintaining stability near or above the zero-shot baseline.

Table 7: **Combined Micro-Dynamics.** Average improvement over Zero-Shot ($\Delta_t$) across the 5-episode duration of a dynamic phase. **Blue** indicates adaptation/learning. **Red** indicates interference/degradation. The **Avg** column shows the mean across all 5 episodes.

| Rule ID | Method | XLand-Minigrid ($\Delta_t$) | | | | | | Kinetix ($\Delta_t$) | | | | | |
|---|---|---|---|---|---|---|---|---|---|---|---|---|---|
| | | **Avg** | Ep 1 | Ep 2 | Ep 3 | Ep 4 | Ep 5 | **Avg** | Ep 1 | Ep 2 | Ep 3 | Ep 4 | Ep 5 |
| Rule 00 | Trad. DR | **-0.025** | 0.000 | -0.024 | -0.031 | -0.035 | -0.037 | **-0.059** | 0.000 | -0.058 | -0.072 | -0.057 | -0.109 |
| | Static | **0.081** | 0.000 | 0.069 | 0.105 | 0.116 | 0.117 | **-0.026** | -0.001 | -0.022 | -0.027 | -0.045 | -0.037 |
| | **Non-Static** | **0.020** | 0.000 | 0.016 | 0.024 | 0.028 | 0.032 | **0.003** | 0.000 | 0.000 | 0.012 | 0.003 | -0.002 |
| Rule 01 | Trad. DR | **-0.110** | -0.143 | -0.122 | -0.106 | -0.094 | -0.085 | **-0.077** | -0.058 | -0.080 | -0.086 | -0.075 | -0.085 |
| | Static | **0.067** | 0.067 | 0.070 | 0.069 | 0.065 | 0.063 | **-0.027** | -0.047 | -0.028 | -0.016 | -0.028 | -0.018 |
| | **Non-Static** | **0.034** | 0.014 | 0.033 | 0.035 | 0.042 | **0.044** | **-0.011** | -0.024 | -0.007 | -0.007 | -0.016 | -0.001 |
| Rule 02 | Trad. DR | **-0.120** | -0.154 | -0.133 | -0.116 | -0.102 | -0.093 | **-0.061** | -0.075 | -0.051 | -0.060 | -0.063 | -0.056 |
| | Static | **0.031** | 0.033 | 0.032 | 0.031 | 0.031 | 0.028 | **-0.036** | -0.045 | -0.036 | -0.027 | -0.035 | -0.035 |
| | **Non-Static** | **0.033** | 0.014 | 0.035 | 0.034 | 0.039 | **0.041** | **0.005** | 0.003 | 0.005 | 0.002 | 0.000 | 0.013 |
| Rule 03 | Trad. DR | **-0.124** | -0.159 | -0.136 | -0.123 | -0.105 | -0.095 | **-0.091** | -0.068 | -0.096 | -0.101 | -0.101 | -0.090 |
| | Static | **0.004** | 0.007 | 0.005 | 0.002 | 0.003 | 0.002 | **-0.019** | 0.005 | -0.030 | -0.023 | -0.026 | -0.020 |
| | **Non-Static** | **0.042** | 0.023 | 0.041 | 0.046 | 0.049 | **0.050** | **-0.004** | 0.021 | -0.010 | -0.011 | -0.011 | -0.009 |
| Rule 04 | Trad. DR | **-0.116** | -0.149 | -0.128 | -0.113 | -0.099 | -0.089 | **-0.098** | -0.145 | -0.097 | -0.071 | -0.089 | -0.086 |
| | Static | **-0.009** | -0.008 | -0.006 | -0.010 | -0.009 | -0.011 | **-0.018** | -0.023 | -0.007 | -0.027 | -0.013 | -0.019 |
| | **Non-Static** | **0.044** | 0.029 | 0.047 | 0.047 | 0.047 | **0.052** | **0.006** | 0.012 | 0.009 | -0.002 | 0.007 | 0.006 |

## C.2 DETAILS OF CONTINUOUS AND STRUCTURED DYNAMICS

In this section, we provide the detailed experimental protocols and comprehensive results for the continuous and structured non-stationary environments.

### C.2.1 DETAILED ENVIRONMENT PROTOCOLS

**Kinetix (Parametric Drift).** We modified the physics engine to dynamically update 7 global parameters: *gravity, friction, motor power, restitution, joint stiffness, motor speed,* and *thruster power.* The parameters evolve based on a normalized time variable $t \in [0, 1]$ via a control function $\alpha_t$. The value of a parameter $p$ at time $t$ is determined by $p_t = p_{\min} + \alpha_t(p_{\max} - p_{\min})$. We define four distinct drift patterns:

- **Linear Drift:** $\alpha_t = t$. Parameters shift monotonically from a start configuration to an end configuration (e.g., friction increases from 0.1 to 1.5).
- **Step Function:** $\alpha_t = \lfloor 8t \rfloor \pmod 2$. Parameters flip between two extremes every $\approx 3.1$ episodes, resulting in 4 full regime switches within the lifetime.
- **Sinusoidal:** $\alpha_t = 0.5(1 + \sin(8\pi t))$. Parameters oscillate smoothly with a frequency of 4 cycles per lifetime (period $\approx 6.25$ episodes).
- **Sawtooth:** $\alpha_t = (4t) \mod 1$. Parameters accumulate gradually and reset abruptly 4 times per lifetime.

**XLand-Minigrid (Rule Evolution).** We designed three structured evolution protocols to test symbolic reasoning stability:

- **Depth Increase:** The reasoning depth of the target object's production rule increases linearly from 0 (direct pickup) to 3 (complex synthesis) over the 25 episodes. This tests the agent's capability to handle increasing cognitive load.
- **Random Cosmetic:** The visual mapping (color/shape IDs) is perturbed continuously. The permutation seed shifts incrementally, testing invariance to symbolic noise.
- **AB Periodic Loop:** Two conflicting rule sets, $\mathcal{R}_A$ and $\mathcal{R}_B$, alternate in a fixed cycle ($\mathcal{R}_A \to \mathcal{R}_B \to \mathcal{R}_A \dots$). Each phase lasts $\approx 3$ episodes, creating high interference pressure where context from $\mathcal{R}_A$ is invalid for $\mathcal{R}_B$.

### C.2.2 COMPREHENSIVE ANALYSIS

We present a combined analysis of macroscopic metrics (Zero-Shot, Sequence Return) and microscopic adaptation dynamics (Episode-wise Improvement). We define $\Delta_{\text{In-Context}} = \text{Seq} - \text{ZS}$ as the overall adaptation gain, and $\Delta_t = R_t - \text{ZS}$ as the instantaneous improvement at episode $t$. We track episodes 6, 12, 18, and 24 to monitor performance at the end of each change cycle.

Table 8 and Table 9 summarize the results. **Training on Static** environments leads to *Negative Transfer* in dynamic settings (indicated by red values). For instance, in the *Kinetix Step* task, the static model's $\Delta_{\text{In-Context}}$ is $-0.013$, and the micro-level analysis shows the degradation worsening over time (Ep 24: $-0.056$). This confirms that the static model treats historical context as invariant truth, leading to catastrophic interference when dynamics shift. In contrast, **Training on Non-Static** environments consistently yields positive calibration (blue values). Even in the challenging *AB Loop* task, where the static model collapses ($\Delta_{24} = -0.085$), the non-stationary model maintains stability ($\Delta_{24} = +0.007$), demonstrating its ability to isolate conflicting contexts and perform robust online adaptation.

Table 8: **Kinetix Results.** Macro Performance (ZS, Seq, $\Delta_{\text{In-Context}}$) and Micro Dynamics ($\Delta_t$ at Ep 6, 12, 18, 24) under various drift patterns.

| Pattern | Training Mode | Macro Metrics | | | Micro Dynamics ($\Delta_t = R_t - \text{ZS}$) | | | |
|---|---|---|---|---|---|---|---|---|
| | | ZS | Seq | $\Delta_{\text{In-Context}}$ | Ep 6 | Ep 12 | Ep 18 | Ep 24 |
| Linear | Static | 0.108 | 0.096 | -0.012 | 0.011 | -0.034 | -0.003 | -0.048 |
| | **Non-Static** | **0.113** | **0.114** | +0.001 | 0.012 | -0.006 | 0.004 | -0.006 |
| Step | Static | 0.087 | 0.074 | -0.013 | -0.015 | -0.029 | -0.043 | -0.056 |
| | **Non-Static** | **0.116** | **0.121** | +0.006 | 0.001 | -0.019 | 0.000 | -0.012 |
| Sinus | Static | 0.103 | 0.096 | -0.007 | -0.001 | -0.006 | -0.037 | -0.024 |
| | **Non-Static** | **0.115** | **0.127** | +0.012 | 0.044 | 0.004 | 0.012 | **0.029** |
| Sawtooth | Static | 0.100 | 0.099 | -0.001 | 0.026 | 0.026 | 0.020 | 0.023 |
| | **Non-Static** | **0.104** | **0.112** | +0.008 | 0.053 | 0.036 | 0.053 | **0.075** |

Table 9: **XLand-Minigrid Results.** Macro Performance and Micro Dynamics under structured rule evolution patterns.

| Pattern | Training Mode | Macro Metrics | | | Micro Dynamics ($\Delta_t = R_t - \text{ZS}$) | | | |
|---|---|---|---|---|---|---|---|---|
| | | ZS | Seq | $\Delta_{\text{In-Context}}$ | Ep 6 | Ep 12 | Ep 18 | Ep 24 |
| Depth | Static | 0.218 | 0.257 | +0.039 | 0.075 | 0.039 | 0.011 | 0.013 |
| | **Non-Static** | **0.303** | **0.315** | +0.012 | 0.015 | 0.015 | 0.009 | 0.004 |
| Cosmetic | Static | 0.148 | 0.162 | +0.014 | 0.036 | 0.006 | -0.006 | -0.007 |
| | **Non-Static** | **0.232** | **0.240** | +0.008 | 0.009 | 0.007 | 0.007 | **0.010** |
| AB Loop | Static | 0.139 | 0.093 | -0.047 | -0.042 | -0.074 | -0.081 | -0.085 |
| | **Non-Static** | **0.223** | **0.225** | +0.003 | 0.002 | -0.001 | 0.003 | **0.007** |

## C.3 COMPARISON WITH META / ONLINE RL

Table 10: **Macro-Performance Comparison.** Zero-Shot (ZS) vs. Sequence (Seq) Return. $\Delta_{\text{In-Context}}$ denotes the adaptation gain (Seq − ZS).

| Method | XLand-Minigrid | | | Kinetix | | |
|---|---|---|---|---|---|---|
| | ZS-Return | Seq-Return | $\Delta_{\text{In-Context}}$ | ZS-Return | Seq-Return | $\Delta_{\text{In-Context}}$ |
| PPO+EWC | 0.221 | 0.202 | -0.019 | -0.217 | -0.276 | -0.059 |
| Online PPO | 0.221 | 0.204 | -0.017 | -0.217 | -0.255 | **-0.038** |
| Static | 0.146 | 0.088 | -0.058 | -0.348 | -0.446 | -0.098 |
| **Non-Static** | **0.224** | **0.240** | **+0.016** | **0.174** | **0.102** | -0.072 |

Table 11: **Combined Micro-Dynamics.** Average improvement over Zero-Shot ($\Delta_t$) across the 5-episode duration of a dynamic phase. **Blue** indicates adaptation/learning. **Red** indicates interference/degradation. The **Avg** column shows the mean across all 5 episodes.

| Rule ID | Method | XLand-Minigrid ($\Delta_t$) | | | | | | Kinetix ($\Delta_t$) | | | | | |
|---|---|---|---|---|---|---|---|---|---|---|---|---|---|
| | | Avg | Ep 1 | Ep 2 | Ep 3 | Ep 4 | Ep 5 | Avg | Ep 1 | Ep 2 | Ep 3 | Ep 4 | Ep 5 |
| Rule 00 | PPO+EWC | **0.006** | 0.000 | 0.024 | 0.012 | -0.031 | 0.025 | **-0.141** | 0.000 | -0.284 | -0.089 | -0.233 | -0.097 |
| | Online PPO | **0.016** | 0.000 | 0.031 | 0.024 | -0.003 | 0.031 | **-0.149** | 0.000 | -0.284 | 0.000 | -0.220 | -0.242 |
| | Static | **0.070** | 0.000 | 0.090 | 0.055 | 0.092 | 0.114 | **-0.103** | 0.000 | -0.088 | -0.097 | -0.161 | -0.169 |
| | **Non-Static** | **0.021** | 0.000 | 0.038 | 0.013 | 0.014 | 0.043 | **-0.232** | 0.000 | -0.245 | -0.303 | -0.260 | -0.352 |
| Rule 01 | PPO+EWC | **-0.057** | -0.060 | -0.051 | -0.064 | -0.054 | -0.055 | **0.038** | -0.021 | -0.038 | 0.034 | 0.056 | 0.158 |
| | Online PPO | **-0.056** | -0.036 | -0.053 | -0.080 | -0.065 | -0.046 | **0.046** | 0.008 | -0.139 | 0.367 | -0.084 | 0.080 |
| | Static | **-0.022** | -0.018 | -0.019 | -0.019 | -0.021 | -0.033 | **-0.002** | -0.007 | 0.081 | 0.020 | -0.071 | -0.032 |
| | **Non-Static** | **0.008** | -0.006 | 0.002 | 0.027 | -0.009 | 0.027 | **-0.056** | 0.121 | -0.056 | -0.095 | -0.070 | -0.183 |
| Rule 02 | PPO+EWC | **-0.006** | -0.015 | -0.013 | 0.031 | -0.005 | -0.029 | **-0.017** | -0.027 | -0.057 | 0.009 | -0.166 | 0.156 |
| | Online PPO | **-0.011** | -0.044 | 0.002 | 0.015 | -0.006 | -0.019 | **0.069** | 0.004 | -0.006 | 0.190 | -0.081 | 0.238 |
| | Static | **-0.110** | -0.107 | -0.117 | -0.096 | -0.106 | -0.124 | **-0.214** | -0.194 | -0.339 | -0.219 | -0.133 | -0.183 |
| | **Non-Static** | **0.017** | -0.012 | 0.028 | 0.032 | 0.005 | 0.031 | **-0.161** | -0.159 | -0.320 | -0.254 | -0.026 | -0.047 |
| Rule 03 | PPO+EWC | **-0.013** | -0.023 | -0.010 | -0.019 | -0.018 | 0.008 | **-0.043** | 0.008 | -0.058 | -0.067 | -0.077 | -0.022 |
| | Online PPO | **-0.014** | -0.011 | -0.018 | 0.013 | -0.021 | -0.032 | **-0.055** | 0.006 | -0.173 | 0.078 | -0.175 | -0.010 |
| | Static | **-0.094** | -0.099 | -0.092 | -0.089 | -0.094 | -0.095 | **-0.155** | -0.049 | -0.269 | -0.168 | -0.154 | -0.135 |
| | **Non-Static** | **0.011** | -0.003 | 0.013 | 0.018 | 0.016 | 0.013 | **0.010** | 0.041 | -0.086 | 0.065 | 0.031 | -0.000 |
| Rule 04 | PPO+EWC | **-0.024** | -0.021 | -0.026 | -0.026 | -0.049 | 0.002 | **-0.133** | -0.127 | -0.163 | -0.052 | -0.245 | -0.077 |
| | Online PPO | **-0.021** | -0.010 | -0.019 | -0.021 | -0.055 | -0.000 | **-0.099** | -0.166 | -0.116 | 0.021 | -0.124 | -0.111 |
| | Static | **-0.137** | -0.131 | -0.141 | -0.137 | -0.138 | -0.136 | **-0.014** | 0.003 | -0.215 | 0.135 | -0.039 | 0.046 |
| | **Non-Static** | **0.020** | -0.001 | 0.014 | 0.038 | 0.009 | 0.042 | **0.078** | 0.055 | 0.237 | 0.006 | -0.040 | 0.132 |

Unlike In-Context RL, which adapts via forward-pass inference, Online RL methods (such as Online PPO and PPO+EWC) require real-time parameter updates via backpropagation during deployment. This necessitates maintaining an independent model instance for each environment, making the massive parallel evaluation used in our main experiments (e.g., 8,192 environments) computationally prohibitive. Therefore, to conduct a feasible comparison, we averaged results across 128 environment instances. To ensure a strong starting point, we initialized the online agents using the pre-trained Domain Randomization (DR) checkpoint. Consistent with our main protocol, the environmental dynamics shift abruptly every 5 episodes, and the agent is not explicitly informed of these change boundaries.

The quantitative results in Table 10 clearly highlight the limitations of gradient-based online adaptation in this setting. As shown in XLand-Minigrid, Online PPO suffers from negative transfer with a $\Delta_{\text{In-Context}}$ of $-0.017$, implying that online gradient updates actually degraded the policy compared to its zero-shot initialization. Even when augmented with EWC regularization, the method fails to achieve positive adaptation ($-0.019$). In Kinetix, a similar trend of degradation is observed, where Online PPO yields a negative gain of $-0.038$. This suggests that gradient updates consis-

tently lag behind the high-frequency environmental shifts, often overfitting to obsolete dynamics or destabilizing the policy.

In stark contrast, our Non-Static CICRL approach demonstrates superior robustness. In XLand-Minigrid, it is the only method to achieve a positive adaptation gain ($+0.016$), effectively utilizing context to improve over the zero-shot baseline. In the complex Kinetix domain, although the adaptation gain metric ($\Delta_{\text{In-Context}}$) is negative for all methods due to the high task difficulty, Non-Static CICRL maintains a significantly higher absolute performance. It attains a Sequence Return of $0.102$, whereas the Static baseline ($-0.446$) and Online PPO ($-0.255$) collapse into negative returns. By leveraging a "frozen weights, fluid context" paradigm, CICRL avoids the instability of rapid gradient updates and maintains viable performance levels where baselines fail.

### C.4 SENSITIVITY TO UNDERLYING RL ALGORITHMS

To evaluate the sensitivity of the CICRL framework to the choice of the underlying RL optimizer, we conduct additional experiments using Reinforce++ under the exact same non-stationary training settings as our main PPO baseline.

Table 12: **PPO vs. Reinforce++.** Comparison of Zero-Shot (ZS) and Sequence (Seq) returns under non-stationary training. $\Delta_{\text{In-Context}}$ denotes the adaptation gain (Seq $-$ ZS). Blue indicates positive adaptation, and red indicates negative interference.

| Algorithm | XLand-Minigrid | | | Kinetix | | |
|---|---|---|---|---|---|---|
| | ZS-Return | Seq-Return | $\Delta_{\text{In-Context}}$ | ZS-Return | Seq-Return | $\Delta_{\text{In-Context}}$ |
| **PPO** | **0.172** | **0.206** | **+0.034** | **0.034** | **0.025** | -0.009 |
| Reinforce++ | 0.019 | 0.017 | -0.002 | -0.134 | -0.141 | -0.007 |

As shown in Table 12, Reinforce++ fails to converge meaningfully in both domains, whereas PPO maintains robust performance. We attribute this performance gap to two inherent challenges in the CICRL setting that demand high algorithmic stability:

- **High Variance in Credit Assignment:** CICRL involves extremely long context windows (up to tens of thousands of steps). Reinforce++, which relies on Monte Carlo sampling, suffers from excessive variance in gradient estimation over such long horizons, making effective credit assignment difficult.
- **Non-stationarity induced Distribution Shift:** Sudden changes in environmental dynamics cause drastic fluctuations in data distribution. PPO's *clipping mechanism* effectively constrains the policy update step size, preventing destructive parameter updates during volatile periods of environmental shift.

This comparison suggests that while sequence models (e.g., Transformers, Gated DeltaNet) provide the representational capacity for in-context adaptation, unlocking this potential requires a stable optimization algorithm capable of handling high variance and distribution shifts.

## D USE OF LARGE LANGUAGE MODELS

In this work, the large language model (LLM) is employed exclusively for text polishing purposes. Its role is limited to refining the linguistic quality, coherence, and stylistic consistency of the textual content, without involvement in data generation, analytical reasoning, or substantive content creation.

