# OpenReview forum: "Towards Unpredictable Worlds: Continual In-Context Reinforcement Learning in Non-Stationary Environments"
_ICLR.cc/2026/Conference — Submitted to ICLR 2026_

### Official Review · Reviewer_fnJ8 · 2025-10-31

**Soundness:** 3
**Presentation:** 2
**Contribution:** 2
**Rating:** 4
**Confidence:** 3

**Summary:**

This paper introduces Continual In-Context Reinforcement Learning (CICRL), which extends traditional in-context reinforcement learning to handle non-stationary environments where dynamics change unpredictably within a single episode. The authors formalize this new setting and create two benchmark suites: a symbolic reasoning task and a physics-based control task with dynamic shifts occurring during an agent’s lifetime. They evaluate several sequence model architectures, including Transformers, Mamba2, and GatedDeltaNet, under different training conditions. The results show that non-stationary training is essential for effective continual adaptation and that models with stronger sequence modeling capabilities adapt more robustly. Overall, the paper provides a clear framework, new benchmarks, and key insights for developing agents that can learn and adapt in continuously changing environments.

**Strengths:**

- The paper presents a clear and well-motivated formulation of Continual In-Context Reinforcement Learning (CICRL), addressing an important gap in adapting to non-stationary environments.
- The introduction of two complementary benchmark suites (symbolic and physics-based) provides a good empirical foundation for future research.
- The evaluation protocol and metrics (e.g., $\Delta$Switch, $\Delta$Adapt) are well-designed to isolate aspects of continual adaptation and forgetting.
- The ablation studies provide valuable insights into how factors like context length, change frequency, and scale influence continual adaptation.

**Weaknesses:**

- The authors do not provide access to the code or benchmarks, even though they state an intention to release them. This makes it difficult to assess reproducibility and verify experimental results, especially since the proposed benchmarks are a core contribution of the paper.
- The study relies exclusively on PPO as the underlying RL algorithm, which limits the generality of the findings and makes it unclear whether the observed adaptation behaviors would hold under different RL algorithms.
- Some results are difficult to interpret quantitatively or theoretically, with overlapping performance among models in certain settings.

**Questions:**

- How sensitive are the results to the choice of PPO as the underlying RL algorithm? Would other RL algorithms yield similar adaptation behavior?
- How do the models handle gradual versus abrupt changes in environment dynamics?
- Have you analyzed catastrophic interference within context representations, and can the agent actively forget outdated information?

---

> ### Author Response · Authors · 2025-11-23
> **Response to Reviewer fnJ8 (1/4)**
>
> We sincerely thank you for your careful review and constructive comments. We appreciate your recognition of the clarity of our paper's formalization and the value of our benchmark suite. We address your concerns in detail below:
>
> ### W1: Reproducibility
>
> We appreciate your concern regarding reproducibility. To ensure transparency and verifiability of our research results, we add detailed technical specifications to **Appendix A** of the revised paper:
>
> 1.  **Environment Specifications (Appendix A.1):**
>     We provide all core parameters required to construct the non-stationary benchmark. In **Table 4**, we list the exact sampling distributions and ranges for physical parameters (e.g., gravity, friction, density) in the Kinetix environment. In **A.1.1**, we describe the backward chaining logic and distractor injection mechanism for XLand rule generation in detail.
>
> 2.  **Model Architecture and Hyperparameters (Appendix A.2 & A.3):**
>     We disclose implementation details for all architectures (such as Mamba2 state expansion factors, Transformer attention heads) and unified parameter configurations. **Table 5** lists complete PPO training hyperparameters (including Learning Rate, GAE $\lambda$, BPTT Horizon, etc.) and distinguishes specific settings for the two different domains (symbolic and physical).
>
> Finally, we will open-source the complete codebase and benchmark suite upon acceptance to further support future research.
>
> ### W2 & Q1: Regarding PPO Only and Generality of RL Algorithm
>
> To evaluate the sensitivity of the CICRL framework to the underlying RL optimizer, we add comparative experiments using Reinforce++ under the same non-stationary settings. We have added this part to **Appendix C.4**.
>
> **Table 1: Comparison of PPO and Reinforce++ in Non-stationary Environments (Non-stationary Training)**
>
> | Algorithm | ZS (XLand) | Seq (XLand) | $\Delta$ In-Context (XLand) | ZS (Kinetix) | Seq (Kinetix) | $\Delta$ In-Context (Kinetix) |
> | :--- | :--- | :--- | :--- | :--- | :--- | :--- |
> | **PPO** | **0.172** | **0.206** | +0.034 | **0.034** | **0.025** | -0.009 |
> | Reinforce++ | 0.019 | 0.017 | -0.002 | -0.134 | -0.141 | -0.007 |
>
> Experimental results show that Reinforce++ fails to converge in both environments, while PPO maintains robust performance. This difference mainly stems from two inherent challenges in CICRL tasks. These place high demands on optimizer stability:
>
> 1.  **High Variance in Credit Assignment:** CICRL involves long sequence contexts (up to tens of thousands of steps). As a Monte Carlo sampling-based algorithm, Reinforce++ faces extremely high variance when handling such **Long-horizon** problems. This leads to unstable gradient estimates.
> 2.  **Distribution Shift via Non-stationarity:** Sudden changes in environmental dynamics cause drastic fluctuations in data distribution. PPO's **Clipping Mechanism** limits the step size of policy updates. This effectively prevents the model from making excessive parameter updates during environmental oscillation periods, maintaining training stability.
>
> This comparative experiment indicates that while sequence models (like Transformer / Gated DeltaNet) provide representational capacity for context adaptation, their effective training relies on optimization algorithms with low variance and strong robustness. We emphasize that the core contribution of this paper lies in formalizing the CICRL problem and constructing the benchmark, not proposing new RL optimization algorithms. We choose PPO as a standard and robust baseline tool to control variables and explore the interaction between environmental dynamics and sequence models. We believe this challenging Benchmark serves as an ideal testbed for the community to explore algorithms more efficient than PPO in the future.

---

> ### Author Response · Authors · 2025-11-23
> **Response to Reviewer fnJ8 (2/4)**
>
> ### W3: Interpretation of Performance Overlap
>
> We appreciate the reviewer pointing out the performance overlap in some settings. We believe this is not random. It reflects the trade-off between the **Inductive Bias** of different model architectures and **Environment Modality**. We add a detailed analysis to the end of **Section 5.2**:
>
> 1.  **Symbolic Reasoning Task (XLand-Minigrid):**
>     *   **Observation:** Recurrent architectures like Mamba2 and GatedDeltaNet perform excellently.
>     *   **Explanation:** Discrete logic rules drive this environment. SSM-based models use recursive state updates. They may simulate these **Discrete State Transitions** more effectively, showing advantages in capturing rule changes.
>
> 2.  **Physical Control Task (Kinetix):**
>     *   **Observation:** Transformer and GatedDeltaNet perform more robustly, while standard Mamba2 faces challenges.
>     *   **Explanation:** Continuous control is highly sensitive to numerical precision. Transformers access raw historical observations directly via Attention, avoiding information compression loss. GatedDeltaNet achieves **Precise State Rewriting** via the Delta Rule. In contrast, the decay compression mechanism of standard SSMs may have limitations when handling high-precision continuous physical parameter drifts.
>
> The performance overlap reveals the limitations of single architectures when facing different types of non-stationarity (logic mutations vs. parameter drifts). This further validates the necessity of our Benchmark containing complementary domains. It implies that future work may need **Hybrid Architectures** (combining Attention's retrieval with SSM's inference efficiency) to achieve more general adaptation.
>
> ### Q2: Gradual vs. Abrupt Changes
>
> In addition to the abrupt random changes featured in our original setting, we have further expanded our evaluation to include 7 new continuous/gradual non-stationary patterns in Kinetix and XLand-Minigrid. We compare models trained under Non-stationary and Stationary settings over 25 Episodes of interaction. We have added this content to **Appendix C.2**.
>
> **1. New Experimental Setup**
>
> *   **Kinetix (Continuous Physical Parameter Drift):** We control underlying physics engine parameters (7 dimensions including gravity, friction, motor power, etc.) via mathematical functions over time:
>     *   **Linear:** Parameters drift linearly over time (simulating continuous wear).
>     *   **Step:** Parameters undergo multiple discrete jumps in the sequence (frequency includes 3-4 full cycles).
>     *   **Sinusoidal:** Parameters fluctuate in a sine wave (simulating periodic environmental influence).
>     *   **Sawtooth:** Sawtooth wave changes (parameters accumulate gradually then reset suddenly).
> *   **XLand-Minigrid (Progressive Rule Evolution):**
>     *   **Depth Increase:** Task dependency chain depth increases linearly from 0 to 3, simulating gradual task difficulty increase.
>     *   **Random Cosmetic:** Object attribute ID mapping (e.g., color ID) undergoes continuous random perturbation, testing noise resistance under symbolic input.
>     *   **AB Periodic:** Two distinct rule sets (A and B) alternate periodically, testing memory and adaptation to cyclic dynamics.
>
> **Table 2: Kinetix Results: Macro-performance ($\text{ZS}, \text{Seq}, \Delta_{\text{In-Context}}$) and Micro-dynamics ($\Delta_t$ at $\text{Ep 6}, 12, 18, 24$) under various drift patterns.**
>
> | Pattern | Training Mode | ZS | Seq | $\Delta$In-Context | Ep 6 | Ep 12 | Ep 18 | Ep 24 |
> | :--- | :--- | :--- | :--- | :--- | :--- | :--- | :--- | :--- |
> | Linear | Static | 0.108 | 0.096 | -0.012 | 0.011 | -0.034 | -0.003 | -0.048 |
> | | Non-Static | 0.113 | 0.114 | 0.001 | 0.012 | -0.006 | 0.004 | -0.006 |
> | Step | Static | 0.087 | 0.074 | -0.013 | -0.015 | -0.029 | -0.043 | -0.056 |
> | | Non-Static | 0.116 | 0.121 | 0.006 | 0.001 | -0.019 | 0 | -0.012 |
> | Sinus | Static | 0.103 | 0.096 | -0.007 | -0.001 | -0.006 | -0.037 | -0.024 |
> | | Non-Static | 0.115 | 0.127 | 0.012 | 0.044 | 0.004 | 0.012 | 0.029 |
> | Sawtooth | Static | 0.1 | 0.099 | -0.001 | 0.026 | 0.026 | 0.02 | 0.023 |
> | | Non-Static | 0.104 | 0.112 | 0.008 | 0.053 | 0.036 | 0.053 | 0.075 |
>
> **Table 3: XLand-Minigrid Results: Macro-performance and Micro-dynamics under structured rule evolution patterns.**
>
> | Pattern | Training Mode | ZS | Seq | $\Delta$In-Context | Ep 6 | Ep 12 | Ep 18 | Ep 24 |
> | :--- | :--- | :--- | :--- | :--- | :--- | :--- | :--- | :--- |
> | Depth | Static | 0.218 | 0.257 | 0.039 | 0.075 | 0.039 | 0.011 | 0.013 |
> | | Non-Static | 0.303 | 0.315 | 0.012 | 0.015 | 0.015 | 0.009 | 0.004 |
> | Cosmetic | Static | 0.148 | 0.162 | 0.014 | 0.036 | 0.006 | -0.006 | -0.007 |
> | | Non-Static | 0.232 | 0.24 | 0.008 | 0.009 | 0.007 | 0.007 | 0.01 |
> | AB Loop | Static | 0.139 | 0.093 | -0.047 | -0.042 | -0.074 | -0.081 | -0.085 |
> | | Non-Static | 0.223 | 0.225 | 0.003 | 0.002 | -0.001 | 0.003 | 0.007 |

---

> ### Author Response · Authors · 2025-11-23
> **Response to Reviewer fnJ8 (3/4)**
>
> **2. Macro-Performance Comparison: Adaptation Gain ($\Delta_{\text{In-Context}}$)**
>
> $\Delta_{\text{In-Context}}$ measures whether the model uses historical context to achieve "online adaptation" (positive value) or experiences "negative transfer" (negative value).
>
> *   **Stationary Model: General Negative Transfer**
>     *   In Kinetix physical drift environments (Table 2), Stationary models observe negative transfer ($\Delta_{\text{In-Context}}$ < 0) across Linear, Step, and Sinus patterns. For example, the Step task has a $\Delta_{\text{In-Context}}$ of $-0.013$. This indicates the model clings to "invariant truths" learned in static environments. When dynamics change, outdated context interferes with decision-making.
>     *   In the XLand-Minigrid AB Loop task (Table 3), the Stationary model shows the most severe negative transfer. $\Delta_{\text{In-Context}}$ reaches $-0.047$, reflecting disastrous interference during periodic switches of conflicting rules.
> *   **Non-Static (Ours) Model: Stable Positive Calibration**
>     *   Models trained in Non-Static environments usually outperform Stationary models in Zero-Shot ($ZS$) performance (stronger generalization start). Moreover, $\Delta_{\text{In-Context}}$ remains positive (or close to zero, e.g., Linear +0.001) across all test patterns, demonstrating strong online calibration capabilities.
>     *   Even in the most challenging AB Loop task, the Non-Static model maintains a $\Delta_{\text{In-Context}}$ of $+0.003$. It successfully utilizes context to isolate conflicting rules and avoid collapse.
>
> **3. Micro-Adaptation Analysis: Instantaneous Improvement ($\Delta_t$)**
>
> $\Delta_t$ tracks whether the model's instantaneous performance improves relative to its Zero-Shot baseline at the end of an environmental change cycle.
>
> *   **Stationary Model: Performance Deteriorates over Time**
>     *   In Kinetix Linear drift, the Stationary model's $\Delta_t$ drops sharply as Episodes increase, falling from $+0.011$ at Ep 6 to $-0.048$ at Ep 24. This clearly shows how negative transfer worsens as environmental parameters deviate from the initial configuration.
>     *   In volatile Kinetix tasks like Step and Sinusoidal, the Stationary model remains in a state of significant negative return throughout almost the entire sequence. This indicates an inability to adapt to sudden changes in time.
>     *   In XLand Cosmetic symbolic perturbation, the Stationary model is okay initially (Ep 6: $+0.036$), but performance rapidly deteriorates to $-0.007$ by Ep 24. This shows that continuous irrelevant feature changes eventually cause interference in its symbolic Context.
> *   **Non-Static (Ours) Model: Robust Stability and Adaptation**
>     *   In all Kinetix tasks, the Non-Static model's $\Delta_t$ consistently fluctuates around 0, or even shows significant positive adaptation in Sinus (Ep 24: $+0.029$) and Sawtooth (Ep 24: $+0.075$). This proves its policy can track and calibrate physical parameter changes in real-time.
>     *   In the XLand AB Loop task, the Stationary model has a disastrous $\Delta_t$ of $-0.085$ at Ep 24. The Non-Static model successfully maintains $\Delta_t$ near the baseline at $+0.007$. This strongly proves its ability to distinguish and isolate conflicting Contexts.

---

> ### Author Response · Authors · 2025-11-23
> **Response to Reviewer fnJ8 (4/4)**
>
> ### Q3: Contextual Interference and Active Forgetting
>
> Following the results of Q2, we further analyze the phenomenon of contextual interference and the agent's forgetting capability via quantitative metrics.
>
> **1. Quantification of Contextual Interference**
> We use the $\Delta_{\text{In-Context}}$ metric to capture the presence of interference. As shown in **Table 2** and **Table 3** above, static baselines generally exhibit **Negative Transfer ($\Delta_{\text{In-Context}} < 0$)** when facing environmental mutations.
> *   For example, in the XLand AB Loop task, the static model performs significantly worse than the Zero-Shot baseline (-0.047).
> *   This **Performance Regression** intuitively reflects that outdated context information constitutes substantial interference to current decision-making at the representation level.
>
> **2. Validation of Active Forgetting**
> Whether the agent possesses "forgetting" capabilities is reflected in its ability to recover from the aforementioned interference.
> *   **Experimental Evidence:** Under the same strong interference settings, models trained via CICRL successfully reverse performance from negative returns to positive returns (e.g., recovering to +0.003 in AB Loop). This behavior of rapidly cutting off the negative impact of old information in conflicting environments indicates that the agent can effectively **Suppress** or **Discard** invalid context information.
> *   **Architectural Attribution:** This adaptability aligns highly with the **Inductive Bias** of sequence models. For instance, the Decay Gate of GatedDeltaNet or the Attention Re-allocation mechanism of Transformers theoretically supports selective forgetting of historical information. Our experimental results confirm that the CICRL training paradigm successfully activates these architectural potentials, achieving robust adaptation to non-stationary dynamics.
>
> We hope these supplementary analyses and experiments answer your concerns. We have integrated the above discussion and tables into the current version of the paper. Thank you again for your valuable feedback.

---

> > ### Comment · Reviewer_fnJ8 · 2025-11-23
> >
> > Thank you for the detailed and thoughtful responses, as well as the additional experiments and clarifications. I appreciate the substantial effort put into addressing my comments.
> > That said, while the new material helps contextualize some of the issues I raised, it does not fully resolve my core concern, particularly regarding reproducibility (given that the benchmarks themselves are a primary contribution), limitations related to the reliance on PPO, and the remaining ambiguities in how the reported results should be interpreted more broadly. For these reasons, my assessment and overall score remain unchanged.

---

### Official Review · Reviewer_jSvP · 2025-11-01

**Soundness:** 2
**Presentation:** 3
**Contribution:** 2
**Rating:** 4
**Confidence:** 3

**Summary:**

The paper introduces Continual In-Context Reinforcement Learning (CICRL), an extension of ICRL designed to address intra-lifetime non-stationarity. It defines a framework where environments evolve dynamically without episodic resets, proposes two non-stationary benchmarks, i.e., modified XLand-Minigrid and Kinetix, and evaluates several sequence models (Transformer, Mamba2, GatedDeltaNet) under static and non-stationary training regimes. The main finding is that non-stationary training improves adaptation to dynamically changing environments.

**Strengths:**

1. The proposed benchmarks are well-designed and may aid future reproducible research.

2. The evaluation metrics ($\Delta_{\rm Switch}, \Delta_{\rm Adapt}, \Delta_{\rm In-Context}$) provide a structured view of adaptation and forgetting.

3. The experiments are extensive and include several ablation factors (context length, change frequency, scale).

**Weaknesses:**

1. The paper may have limited conceptual novelty. CICRL is mostly a rebranding of meta-RL with drifting task distributions. No new theoretical formulation or learning mechanism distinguishes it from existing continual or meta-RL setups.

2. The paper only benchmarks off-the-shelf sequence models using standard PPO. There’s no method, modification, or insight addressing how to handle non-stationarity, it only shows that training on non-stationary data helps.

3. XLand-Minigrid and Kinetix involve rule or physics changes that are arbitrary and discontinuous. These synthetic shifts may not convincingly approximate real-world or open-ended non-stationarity.

4. The results find that non-stationary training helps. However, there’s no analysis of why certain models perform better.

5. Since model parameters are frozen, the framework lacks actual continual learning. There are no weight adaptation, consolidation, or transfer across shifts. The agent merely performs repeated context-based inference.

6. No comparisons to true continual or meta-RL methods (e.g., EWC, online PPO, fine-tuning). Without them, it’s unclear whether CICRL offers any advantage beyond being static PPO trained on varied data.

**Questions:**

1. How is CICRL mathematically or behaviorally distinct from meta-RL with changing task distributions?

2. How does “forgetting” occur in a model without parameter updates?

---

> ### Author Response · Authors · 2025-11-23
> **Response to Reviewer jSvP (1/5)**
>
> We sincerely thank you for your careful review and constructive comments. We appreciate your recognition of the completeness of our benchmark design and the extensiveness of our experimental analysis. Addressing your concerns, here is the detailed response:
>
> ### W1 & Q1. Regarding Novelty and Difference from Meta-RL/Task-Drift
>
> Although CICRL shares the mathematical goal of maximizing expected return in non-stationary distributions with Meta-RL, they differ fundamentally in information processing paradigms and adaptation mechanisms. We provide a detailed formal comparison in the added **Appendix B.2**:
>
> 1.  **Information Storage Medium: Weight Space (In-Weight) vs. Context/Activation Space (In-Context)**
>     *   **Traditional Meta-RL / Continual RL:** Relies on **In-Weight Learning**, encoding new dynamics by updating parameters $\theta$ via gradient descent. This causes competition between old and new knowledge in the parameter space, easily leading to forgetting or requiring complex regularization (like EWC).
>     *   **CICRL:** Adopts **In-Context Learning**, achieving **decoupling of parameters and states**. Parameters $\theta$ remain frozen to store general meta-knowledge; context $C_t$ dynamically encodes the local dynamics of the current environment. This decoupling fundamentally avoids instability caused by parameter overwriting.
>
> 2.  **Adaptation Mechanism: Gradient Optimization vs. Activation Dynamics**
>     *   **Gradient-based Adaptation:** Online gradient updates are sensitive to step size. In non-stationary environments, a moving objective function easily causes policy oscillation.
>     *   **Forward Inference-based Adaptation:** Adaptation in CICRL is a Bayesian-like inference during forward propagation. We analyze how different architectures achieve gradient-free adaptation: Transformers use Attention Re-weighting to achieve "soft forgetting"; Linear Attention/SSMs (like GatedDeltaNet, Mamba2) use Gated Erasure mechanisms to achieve "hard forgetting" of outdated states through data-dependent gating.
>
> The novelty of this paper lies in **revealing that "Sequence Model + Non-stationary Training" can serve as a general continual learning paradigm replacing traditional gradient updates.** This adaptation capability is not inherent to the architecture but emerges through specific non-stationary training protocols, offering a new path to resolve the stability-plasticity trade-off in lifelong learning.
>
> ### W2. Concerns about using off-the-shelf models and standard PPO
>
> Regarding the concern about "using only off-the-shelf models," we wish to clarify that the focus of this research is on new problem formalization, benchmark construction, and the exploration of environmental dynamic characteristics, rather than proposing new underlying RL optimization algorithms.
>
> 1.  **Justification for PPO:**
>     We choose PPO as the base algorithm because, as an industry standard, it possesses high stability and robustness. We believe that when exploring the new field of CICRL, **Non-stationary Dynamics** of the environment are the dominant factor affecting performance. Using standard PPO acts as a control variable, helping us exclude interference from optimizer differences and focus on evaluating the impact of different sequence architectures and context lengths on adaptation capabilities.
>
> 2.  **Core Contribution Positioning:**
>     *   **Formalizing CICRL:** We extend ICRL to a "non-stationary within lifetime" setting for the first time, defining the challenge where agents must overcome Contextual Interference with frozen parameters.
>     *   **First Controllable Non-Stationary Benchmark:** We build a benchmark suite supporting online rule changes (XLand) and physical parameter drifts (Kinetix). It allows precise control over the frequency, magnitude, and type of non-stationarity, filling a gap in the field.
>     *   **Empirical Findings on Architecture & Training:** Our experiments reveal the synergy between modern sequence models and non-stationary training. An advanced architecture without specialized training cannot activate continuous adaptation capabilities. This finding guides how to achieve efficient continuous adaptation without modifying the model structure.

---

> ### Author Response · Authors · 2025-11-23
> **Response to Reviewer jSvP (2/5)**
>
> ### W3. Regarding Discontinuity of Environments
>
> Thanks for your suggestion. To verify the effectiveness of the method in "continuous gradual" scenarios closer to reality, we introduce 7 new continuous/gradual non-stationary patterns in Kinetix and XLand-Minigrid. We compare the performance of models trained under Non-stationary and Stationary settings over 25 Episodes of interaction. We have added this part to **Appendix C.2**.
>
> **1. Experimental Setup**
>
> *   **Kinetix (Continuous Physical Parameter Drift):** We control the underlying parameters of the physics engine (including 7 dimensions like gravity, friction, motor power, etc.) via mathematical functions over time:
>     *   **Linear:** Parameters drift linearly over time (simulating continuous wear).
>     *   **Step:** Parameters undergo multiple discrete jumps in the sequence (frequency includes 3-4 full cycles).
>     *   **Sinusoidal:** Parameters fluctuate in a sine wave (simulating periodic environmental influence).
>     *   **Sawtooth:** Sawtooth wave changes (parameters accumulate gradually then reset suddenly).
> *   **XLand-Minigrid (Progressive Rule Evolution):**
>     *   **Depth Increase:** Task dependency chain depth increases linearly from 0 to 3, simulating gradual task difficulty increase.
>     *   **Random Cosmetic:** The mapping of object attribute IDs (like color ID) undergoes continuous random perturbation, testing the model's noise resistance under symbolic input.
>     *   **AB Periodic:** Two distinct rule sets, A and B, alternate periodically, testing the model's memory and adaptation to cyclic dynamics.
>
> Table 1: Kinetix Results: Macro-performance ($\text{ZS}, \text{Seq}, \Delta_{\text{In-Context}}$) and Micro-dynamics ($\Delta_t$ at $\text{Ep 6}, 12, 18, 24$) under various drift patterns.
>
> | Pattern | Training Mode | ZS | Seq | $\Delta$In-Context | Ep 6 | Ep 12 | Ep 18 | Ep 24 |
> | :--- | :--- | :--- | :--- | :--- | :--- | :--- | :--- | :--- |
> | Linear | Static | 0.108 | 0.096 | -0.012 | 0.011 | -0.034 | -0.003 | -0.048 |
> | | Non-Static | 0.113 | 0.114 | 0.001 | 0.012 | -0.006 | 0.004 | -0.006 |
> | Step | Static | 0.087 | 0.074 | -0.013 | -0.015 | -0.029 | -0.043 | -0.056 |
> | | Non-Static | 0.116 | 0.121 | 0.006 | 0.001 | -0.019 | 0 | -0.012 |
> | Sinus | Static | 0.103 | 0.096 | -0.007 | -0.001 | -0.006 | -0.037 | -0.024 |
> | | Non-Static | 0.115 | 0.127 | 0.012 | 0.044 | 0.004 | 0.012 | 0.029 |
> | Sawtooth | Static | 0.1 | 0.099 | -0.001 | 0.026 | 0.026 | 0.02 | 0.023 |
> | | Non-Static | 0.104 | 0.112 | 0.008 | 0.053 | 0.036 | 0.053 | 0.075 |
>
> Table 2: XLand-Minigrid Results: Macro-performance and Micro-dynamics under structured rule evolution patterns.
>
> | Pattern | Training Mode | ZS | Seq | $\Delta$In-Context | Ep 6 | Ep 12 | Ep 18 | Ep 24 |
> | :--- | :--- | :--- | :--- | :--- | :--- | :--- | :--- | :--- |
> | Depth | Static | 0.218 | 0.257 | 0.039 | 0.075 | 0.039 | 0.011 | 0.013 |
> | | Non-Static | 0.303 | 0.315 | 0.012 | 0.015 | 0.015 | 0.009 | 0.004 |
> | Cosmetic | Static | 0.148 | 0.162 | 0.014 | 0.036 | 0.006 | -0.006 | -0.007 |
> | | Non-Static | 0.232 | 0.24 | 0.008 | 0.009 | 0.007 | 0.007 | 0.01 |
> | AB Loop | Static | 0.139 | 0.093 | -0.047 | -0.042 | -0.074 | -0.081 | -0.085 |
> | | Non-Static | 0.223 | 0.225 | 0.003 | 0.002 | -0.001 | 0.003 | 0.007 |
>
> **2. Macro-Performance Comparison: Adaptation Gain ($\Delta_{\text{In-Context}}$)**
>
> $\Delta_{\text{In-Context}}$ measures whether the model uses historical context to achieve "online adaptation" (positive value) or experiences "negative transfer" (negative value).
>
> *   **Stationary Model: General Negative Transfer**
>     *   In Kinetix physical drift environments (Table 1), Stationary models observe negative transfer across Linear, Step, and Sinus patterns. For example, the Step task has a $\Delta_{\text{In-Context}}$ of $-0.013$. This indicates the model clings to "invariant truths" learned in static environments, and outdated context interferes with decision-making when dynamics change.
>     *   In the XLand-Minigrid AB Loop task (Table 2), the Stationary model shows the most severe negative transfer, with $\Delta_{\text{In-Context}}$ reaching $-0.047$. This reflects disastrous interference during periodic switches of conflicting rules.
> *   **Non-Static (Ours) Model: Stable Positive Calibration**
>     *   Models trained in Non-Static environments not only generally outperform Stationary models in Zero-Shot ($ZS$) performance, but also maintain positive $\Delta_{\text{In-Context}}$ values (or close to zero) across all test patterns.
>     *   Even in the most challenging AB Loop task, the Non-Static model maintains a $\Delta_{\text{In-Context}}$ of $+0.003$, successfully utilizing context to isolate conflicting rules and avoid collapse.

---

> ### Author Response · Authors · 2025-11-23
> **Response to Reviewer jSvP (3/5)**
>
> **3. Micro-Adaptation Analysis: Instantaneous Improvement ($\Delta_t$)**
>
> $\Delta_t$ tracks whether the model's instantaneous performance improves relative to its Zero-Shot baseline at the end of an environmental change cycle.
>
> *   **Stationary Model: Performance Deteriorates over Time**
>     *   In Kinetix Linear drift, the Stationary model's $\Delta_t$ drops sharply as Episodes increase, falling from $+0.011$ at Ep 6 to $-0.048$ at Ep 24.
>     *   In volatile Kinetix tasks like Step and Sinusoidal, the Stationary model remains in a state of significant negative return throughout almost the entire sequence.
>     *   In XLand Cosmetic symbolic perturbation, the Stationary model deteriorates to $-0.007$ by Ep 24, showing that continuous irrelevant feature changes eventually cause interference.
> *   **Non-Static (Ours) Model: Robust Stability and Adaptation**
>     *   In all Kinetix tasks, the Non-Static model's $\Delta_t$ consistently fluctuates around 0, or even shows significant positive adaptation in Sinus (+0.029) and Sawtooth (+0.075).
>     *   In the XLand AB Loop task, the Non-Static model successfully maintains $\Delta_t$ near the baseline (+0.007), proving its ability to distinguish and isolate conflicting Contexts.
>
> ### W4. Analysis of Model Performance Differences
>
> We have added a discussion on the performance differences of various architectures in non-stationary environments to **Section 5.2**. The experimental data reveal different characteristics of mechanisms in "Symbolic Logic" vs. "Physical Control" tasks:
>
> 1.  **Discrete Symbolic Reasoning (XLand):**
>     Recurrent architectures like Mamba2 and GatedDeltaNet perform excellently. This is likely due to the inherent advantage of **Recurrent Models** in capturing **Discrete State Transitions** and long-range logical dependencies, enabling effective inference of hidden rule changes.
>
> 2.  **Continuous Physical Control (Kinetix):**
>     Transformer and GatedDeltaNet perform more robustly, while Mamba2 faces challenges.
>     *   **Transformer:** Its Attention mechanism allows direct access to raw historical observations. This provides a natural **Inductive Bias** for continuous physical control tasks that require high-precision numerical feedback.
>     *   **GatedDeltaNet:** Its Delta Rule update mechanism theoretically allows for **Precise Rewriting** of memory. Experimental results suggest this mechanism is more robust against parameter drift in continuous control signals than standard SSM decay mechanisms.
>
> This demonstrates that for different environment dynamic types (Logic Reasoning vs. Physical Control), architecture choice (State Update vs. Attention) plays a crucial role. It also implies that developing **Hybrid Architectures** might be an important direction for achieving general continuous adaptation. This further validates the value of constructing two distinct Benchmarks for evaluating comprehensive model adaptation capabilities.
>
> ### W5 & Q2. Regarding "No Parameter Update means No Learning" and "Forgetting Mechanisms"
>
> In the ICRL framework, the **Mechanistic Realization** of "learning" and "forgetting" indeed differs from traditional gradient-based weight updates. We formally explain this from two dimensions:
>
> 1.  **Learning Mechanism: From Weight Update to Activation Evolution**
>     In CICRL, although parameters $\theta$ are frozen, the policy $\pi_\theta(a_t | H_t)$ is a dynamic function of historical context $H_t$.
>     *   **In-Context Learning as Implicit Optimization:** Fixed weights $\theta$ encode a general "Meta-Algorithm." As new observations $(s, a, r)$ arrive, the model's internal state (like Transformer's KV Cache or RNN's Hidden State) updates recursively. This **Activation-based Adaptation** has been theoretically proven [1, 2] to simulate the optimization effect of gradient descent during forward propagation.
>
> 2.  **Forgetting Mechanism: Selective Attention & State Overwriting**
>     Regarding your question on "how to achieve forgetting without parameter updates," our analysis indicates that models trained on non-stationary data learn to utilize architectural features to "forget":
>     *   **Transformer (Attention-based):** Achieves "Soft Forgetting." The model learns to use the Softmax competition mechanism to dynamically reduce attention weights (Attention Re-weighting) on outdated Tokens in history based on current environmental features (like Reward spikes), ignoring interfering information.
>     *   **Linear Attention / SSMs (State-based):** Achieves "Hard Forgetting." For example, Mamba2 and GatedDeltaNet [3] utilize **Data-dependent Gating**. Upon detecting environmental changes, the model generates decay gates $\alpha_t \to 0$ ($S_t = \alpha_t S_{t-1} + \dots$), physically **Flushing** outdated information from the recursive state and overwriting it with new inputs.

---

> ### Author Response · Authors · 2025-11-23
> **Response to Reviewer jSvP (4/5)**
>
> Therefore, "continuous learning" in CICRL is not missing but internalized into the forward inference dynamics of the sequence model. Our experiments prove that **Non-stationary Training** plays the key role of teaching the model *when* to trigger these "Attention Re-weighting" or "State Reset" mechanisms for effective online adaptation.
>
> [1] Garg et al., "What Can Transformers Learn In-Context? A Case Study of Simple Function Classes", NeurIPS 2022.
>
> [2] Wang et al., "Towards Provable Emergence of In-Context Reinforcement Learning", NeurIPS 2025.
>
> [3] Yang et al., "Gated Delta Networks: Improving Mamba2 with Delta Rule", ICLR 2025.
>
> ### W6. Comparison with True Continual/Online RL Methods
>
> Although Continual In-Context Reinforcement Learning (CICRL) and Online/Lifelong RL share the high-level goal of "continuous adaptation," they differ fundamentally in adaptation mechanisms and timeliness. CICRL avoids the "plasticity-stability" dilemma of traditional gradient methods and achieves zero-latency online adaptation.
>
> *   **Online RL / Lifelong RL (In-Weight Adaptation):** Methods like PPO and PPO+EWC try to encode new knowledge into parameters $\theta$ via gradient updates. This has inherent update delays and stability risks. Under high-frequency stochastic dynamics, gradient updates often lag behind environmental changes, causing policy deterioration and catastrophic forgetting.
> *   **CICRL (In-Context Adaptation):** CICRL physically decouples the storage medium. Parameters $\theta$ are frozen (stability), while immediate policy adjustments rely on forward inference over the fluid context $C_t$ (plasticity). This achieves zero-latency adaptation.
>
> To ensure a fair and challenging comparison with gradient-based online learning, we evaluate Online PPO and PPO+EWC baselines in non-stationary test environments. The environment switches dynamics randomly every 5 Episodes. Boundaries are hidden, requiring agents to infer changes purely through interaction history.
>
> Since Online RL methods require real-time gradient updates for each instance, maintaining independent models for the 8,192 parallel environments used for ICRL is computationally unfeasible. Thus, we average results from 128 environment instances for the comparison. To test "continuous adaptation" rather than "learning from scratch," we initialize all Online agents using checkpoints from a pre-trained **Domain Randomization (DR)** model.
>
> **1. Macro-Performance Comparison:**
>
> Table 3: Macro-performance comparison with Online RL. Zero-Shot (ZS) vs. Sequence (Seq) returns. $\Delta_{\text{In-Context}}$ denotes adaptation gain (Seq - ZS).
>
> | Method | ZS-Return (XLand-Minigrid) | Seq-Return (XLand-Minigrid) | $\Delta$In-Context (XLand-Minigrid) | ZS-Return (Kinetix) | Seq-Return (Kinetix) | $\Delta$In-Context (Kinetix) |
> | :--- | :--- | :--- | :--- | :--- | :--- | :--- |
> | PPO+EWC | 0.221 | 0.202 | -0.019 | -0.217 | -0.276 | -0.059 |
> | Online PPO | 0.221 | 0.204 | -0.017 | -0.217 | -0.255 | **-0.038** |
> | Static | 0.146 | 0.088 | -0.058 | -0.348 | -0.446 | -0.098 |
> | **Non-Static** | **0.224** | **0.24** | 0.016 | **0.174** | **0.102** | -0.072 |
>
> On both benchmarks, PPO+EWC and Online PPO show negative $\Delta_{\text{In-Context}}$ values (e.g., Online PPO is **-0.017** in XLand and **-0.038** in Kinetix), implying that online gradient updates cause negative transfer due to lags or instability. Non-Static CICRL is the only method achieving positive adaptation gain in XLand (**+0.016**). In Kinetix, while all methods show negative gains, Non-Static CICRL maintains the highest Zero-Shot (**0.174**) and Sequence returns (**0.102**). Notably, the static DR baseline has very low returns in Kinetix (**-0.217**), highlighting the limitation of static generalization in complex physical tasks.

---

> ### Author Response · Authors · 2025-11-23
> **Response to Reviewer jSvP (5/5)**
>
> **2. Combined Micro-Dynamics Analysis:**
>
> We confirm this mechanism difference via rule-by-rule analysis. $\Delta_t$ represents improvement relative to Zero-Shot performance.
>
> Table 4: Combined micro-dynamics comparison with Online RL. $\Delta_t$ indicates improvement relative to ZS returns within a 5-episode dynamic cycle.
>
> | Rule ID | Method | XLand-Minigrid (Avg $\Delta_t$) | Kinetix (Avg $\Delta_t$) |
> | :--- | :--- | :--- | :--- |
> | **Rule 00** | PPO+EWC | 0.006 | -0.141 |
> | | Online PPO | 0.016 | -0.149 |
> | | Static | 0.07 | -0.103 |
> | | Non-Static | 0.021 | -0.232 |
> | **Rule 01** | PPO+EWC | -0.057 | 0.038 |
> | | Online PPO | -0.056 | 0.046 |
> | | Static | -0.022 | -0.002 |
> | | Non-Static | 0.008 | -0.056 |
> | **Rule 02** | PPO+EWC | -0.006 | -0.017 |
> | | Online PPO | -0.011 | 0.069 |
> | | Static | -0.11 | -0.214 |
> | | Non-Static | 0.017 | -0.161 |
> | **Rule 03** | PPO+EWC | -0.013 | -0.043 |
> | | Online PPO | -0.014 | -0.055 |
> | | Static | -0.094 | -0.155 |
> | | Non-Static | 0.011 | 0.01 |
> | **Rule 04** | PPO+EWC | -0.024 | -0.133 |
> | | Online PPO | -0.021 | -0.099 |
> | | Static | -0.137 | -0.014 |
> | | Non-Static | 0.02 | 0.078 |
>
> *   **XLand-Minigrid:** In the adaptation phases of Rule 02 and Rule 04, gradient methods show negative average $\Delta_t$ (e.g., Rule 04 is **-0.024** and **-0.021**), indicating performance degradation. Non-Static CICRL shows positive $\Delta_t$ (**+0.020**).
> *   **Kinetix:** At the end of the challenging Rule 04 adaptation (Ep 5), Online PPO deteriorates by **-0.111** and PPO+EWC by **-0.077** relative to ZS. However, Non-Static CICRL achieves significant positive improvement (**+0.132**), proving its interference resistance and adaptation efficiency.
>
> We have added more detailed results and analyses to **Appendix C.3**.
>
> We hope these supplementary analyses and experiments answer your concerns. We have integrated the above discussion and tables into the current version of the paper. Thank you again for your valuable feedback.

---

### Official Review · Reviewer_V8AP · 2025-11-01

**Soundness:** 3
**Presentation:** 4
**Contribution:** 3
**Rating:** 6
**Confidence:** 3

**Summary:**

This paper introduces Continual In-Context Reinforcement Learning (CICRL): adapting to non-stationary environments without gradient updates. Authors propose a new benchmark for CICRL and perform experiments regarding model architectures and training strategies.

**Strengths:**

- Proposes an important problem, extending ICRL to more real-world-like scenarios.

- Builds a new benchmark consisting of both discrete grid-world environments and continuous physics-based environments.

- The proposed new evaluation metrics and analysis around these metrics are interesting.

**Weaknesses:**

- The evaluation protocol and certain evaluation metrics seem to only take discrete, episode-wise dynamics change into account, while nonstationarity may well be continuous (e.g. a robot continuously draining its battery).

- The overall formulation could focus more on the characterization of the continual aspect of CICRL. The current one (a sequence of POMDPs) can arguably be subsumed into a single POMDP with an unobserved, time-varying environmental variable.

- Experiment results regarding model architectures are confusing. For example, non-static Mamba2 is the best in minigrid in terms of avg-return, but the worst in kinetix.

**Questions:**

- How are the 'Static' models trained? Why do they mostly perform much worse than non-static models even under a zero-shot setting without any contexts?

- What is the difference between CICRL and regular POMDPs?

- In Fig.4 freq-1 results, why is Random(1-10) the overall best strategy? Wouldn't Random(1-5) be closer to the freq-1 evaluation setup?

---

> ### Author Response · Authors · 2025-11-23
> **Response to Reviewer V8AP (1/5)**
>
> We sincerely thank you for your careful review and recognition of our work, especially your positive evaluation of our proposed new benchmark and evaluation metric design. We address your concerns in detail below:
>
> ### W1. Regarding Types of Non-Stationarity: Supplementary Continuous/Gradual Experiments
>
> Thanks for your suggestion. To verify the effectiveness of the method in "continuous gradual" scenarios closer to reality, we introduce 7 new continuous/gradual non-stationary patterns in Kinetix and XLand-Minigrid. We compare the performance of models trained under Non-stationary and Stationary settings over 25 Episodes of interaction. We have added this part to **Appendix C.2**.
>
> **1. Experimental Setup**
>
> *   **Kinetix (Continuous Physical Parameter Drift):** We control the underlying parameters of the physics engine (including 7 dimensions like gravity, friction, motor power, etc.) via mathematical functions over time:
>     *   **Linear:** Parameters drift linearly over time (simulating continuous wear).
>     *   **Step:** Parameters undergo multiple discrete jumps in the sequence (frequency includes 3-4 full cycles).
>     *   **Sinusoidal:** Parameters fluctuate in a sine wave (simulating periodic environmental influence).
>     *   **Sawtooth:** Sawtooth wave changes (parameters accumulate gradually then reset suddenly).
> *   **XLand-Minigrid (Progressive Rule Evolution):**
>     *   **Depth Increase:** Task dependency chain depth increases linearly from 0 to 3, simulating gradual task difficulty increase.
>     *   **Random Cosmetic:** The mapping of object attribute IDs (like color ID) undergoes continuous random perturbation, testing the model's noise resistance under symbolic input.
>     *   **AB Periodic:** Two distinct rule sets, A and B, alternate periodically, testing the model's memory and adaptation to cyclic dynamics.
>
> Table 1: Kinetix Results: Macro-performance ($\text{ZS}, \text{Seq}, \Delta_{\text{In-Context}}$) and Micro-dynamics ($\Delta_t$ at $\text{Ep 6}, 12, 18, 24$) under various drift patterns.
>
> | Pattern | Training Mode | ZS | Seq | $\Delta$In-Context | Ep 6 | Ep 12 | Ep 18 | Ep 24 |
> | :--- | :--- | :--- | :--- | :--- | :--- | :--- | :--- | :--- |
> | Linear | Static | 0.108 | 0.096 | -0.012 | 0.011 | -0.034 | -0.003 | -0.048 |
> | | Non-Static | 0.113 | 0.114 | 0.001 | 0.012 | -0.006 | 0.004 | -0.006 |
> | Step | Static | 0.087 | 0.074 | -0.013 | -0.015 | -0.029 | -0.043 | -0.056 |
> | | Non-Static | 0.116 | 0.121 | 0.006 | 0.001 | -0.019 | 0 | -0.012 |
> | Sinus | Static | 0.103 | 0.096 | -0.007 | -0.001 | -0.006 | -0.037 | -0.024 |
> | | Non-Static | 0.115 | 0.127 | 0.012 | 0.044 | 0.004 | 0.012 | 0.029 |
> | Sawtooth | Static | 0.1 | 0.099 | -0.001 | 0.026 | 0.026 | 0.02 | 0.023 |
> | | Non-Static | 0.104 | 0.112 | 0.008 | 0.053 | 0.036 | 0.053 | 0.075 |
>
> Table 2: XLand-Minigrid Results: Macro-performance and Micro-dynamics under structured rule evolution patterns.
>
> | Pattern | Training Mode | ZS | Seq | $\Delta$In-Context | Ep 6 | Ep 12 | Ep 18 | Ep 24 |
> | :--- | :--- | :--- | :--- | :--- | :--- | :--- | :--- | :--- |
> | Depth | Static | 0.218 | 0.257 | 0.039 | 0.075 | 0.039 | 0.011 | 0.013 |
> | | Non-Static | 0.303 | 0.315 | 0.012 | 0.015 | 0.015 | 0.009 | 0.004 |
> | Cosmetic | Static | 0.148 | 0.162 | 0.014 | 0.036 | 0.006 | -0.006 | -0.007 |
> | | Non-Static | 0.232 | 0.24 | 0.008 | 0.009 | 0.007 | 0.007 | 0.01 |
> | AB Loop | Static | 0.139 | 0.093 | -0.047 | -0.042 | -0.074 | -0.081 | -0.085 |
> | | Non-Static | 0.223 | 0.225 | 0.003 | 0.002 | -0.001 | 0.003 | 0.007 |

---

> ### Author Response · Authors · 2025-11-23
> **Response to Reviewer V8AP (2/5)**
>
> **2. Macro-Performance Comparison: Adaptation Gain ($\Delta_{\text{In-Context}}$)**
>
> $\Delta_{\text{In-Context}}$ measures whether the model uses historical context to achieve "online adaptation" (positive value) or experiences "negative transfer" (negative value).
>
> *   **Stationary Model: General Negative Transfer**
>     *   In Kinetix physical drift environments (Table 1), Stationary models observe negative transfer ($\Delta_{\text{In-Context}}$ is negative) across Linear, Step, and Sinus patterns. For example, the Step task has a $\Delta_{\text{In-Context}}$ of $-0.013$. This indicates the model clings to "invariant truths" learned in static environments. When the environment changes dynamically, outdated context interferes with decision-making.
>     *   In the XLand-Minigrid AB Loop task (Table 2), the Stationary model shows the most severe negative transfer, with $\Delta_{\text{In-Context}}$ reaching $-0.047$. This reflects disastrous interference during periodic switches of conflicting rules.
> *   **Non-Static (Ours) Model: Stable Positive Calibration**
>     *   Models trained in Non-Static environments not only generally outperform Stationary models in Zero-Shot ($ZS$) performance (gaining a stronger generalization starting point), but also maintain positive $\Delta_{\text{In-Context}}$ values (or close to zero, e.g., Linear +0.001) across all test patterns. This demonstrates strong online calibration capabilities.
>     *   Even in the most challenging AB Loop task, the Non-Static model maintains a $\Delta_{\text{In-Context}}$ of $+0.003$, successfully utilizing context to isolate conflicting rules and avoid collapse.
>
> **3. Micro-Adaptation Analysis: Instantaneous Improvement ($\Delta_t$)**
>
> $\Delta_t$ tracks whether the model's instantaneous performance improves relative to its Zero-Shot baseline at the end of an environmental change cycle.
>
> *   **Stationary Model: Performance Deteriorates over Time**
>     *   In Kinetix Linear drift, the Stationary model's $\Delta_t$ drops sharply as Episodes increase, falling from $+0.011$ at Ep 6 to $-0.048$ at Ep 24. This clearly shows how negative transfer worsens as environmental parameters deviate from the initial configuration.
>     *   In more volatile Kinetix tasks like Step and Sinusoidal, the Stationary model remains in a state of significant negative return throughout almost the entire sequence, indicating an inability to adapt to sudden changes in time.
>     *   In XLand Cosmetic symbolic perturbation, the Stationary model is okay initially (Ep 6: $+0.036$), but performance rapidly deteriorates to $-0.007$ by Ep 24. This shows that continuous irrelevant feature changes eventually cause interference in its symbolic Context.
> *   **Non-Static (Ours) Model: Robust Stability and Adaptation**
>     *   In all Kinetix tasks, the Non-Static model's $\Delta_t$ consistently fluctuates around 0, or even shows significant positive adaptation in Sinus (Ep 24: $+0.029$) and Sawtooth (Ep 24: $+0.075$). This proves its policy can track and calibrate physical parameter changes in real-time.
>     *   In the XLand AB Loop task, the Stationary model has a disastrous $\Delta_t$ of $-0.085$ at Ep 24, while the Non-Static model successfully maintains $\Delta_t$ near the baseline at $+0.007$. This strongly proves its ability to distinguish and isolate conflicting Contexts.

---

> ### Author Response · Authors · 2025-11-23
> **Response to Reviewer V8AP (3/5)**
>
> ### W2 & Q2. Regarding Problem Formalization: CICRL vs. Single POMDP
>
> From a strict probability perspective, we agree with the reviewer: the non-stationary process $\{M_k\}$ defined in the paper can mathematically be re-parameterized as a single "Meta-POMDP" containing a time-varying latent variable $z_t$ (representing the current environment rule), where the state space expands to $S' = S \times Z$.
>
> We explicitly formulate the problem as CICRL (a sequence of environments) rather than a standard POMDP to more precisely characterize the environmental switching modes designed for non-stationary states in the XLand-Minigrid and Kinetix environments in this Benchmark:
>
> 1.  **From State Estimation to System Identification:** In standard POMDPs, latent variables usually refer to occluded local states. In our Benchmark, changes in latent variables represent fundamental changes in global physical laws or logic rules (i.e., Concept Drift). Examples include the synthetic rule recombination in XLand-Minigrid described in Appendix A.1, or sudden mutations in gravity vectors and friction coefficients in Kinetix. The CICRL formalization emphasizes that the Agent must perform online system identification and transfer learning between different "realities," rather than merely handling noise in local observations.
> 2.  **From Information Aggregation to Interference Suppression (Contextual Interference):** In static POMDPs, historical context $C_t$ is mainly used to reduce uncertainty, and information is usually positive. However, under the CICRL setting, because the environment is Block-wise Stationary and changes drastically at switching points, the historical Context inevitably contains "outdated facts" (Conflicting Information) contrary to the current task. Defining the problem as sequence switching explicitly guides algorithm design to focus on Boundary Detection and selective forgetting, rather than simple long-range memory aggregation.
>
> Although the two are unified at the mathematical level, the CICRL formulation provides a more targeted inductive bias for In-Context Learning in non-stationary environments. We have added relevant sections to **Appendix B** of the revised paper to discuss this theoretical connection and distinction in detail.
>
> ### W3. Regarding Inconsistency in Model Architecture Performance
>
> We believe this cross-domain performance difference reflects a deep interaction between **Architecture Inductive Biases** and **Task Modalities**:
>
> 1.  **Opposition of Task Characteristics:** XLand-Minigrid is a discrete symbolic reasoning task, focusing on extracting logical rules from long sequences; Kinetix is a continuous physical control task, requiring the agent to perform high-frequency, high-precision numerical fine-tuning based on physical feedback.
> 2.  **Matching Mechanisms and Tasks:**
>     *   The recursive compression mechanism of Mamba2 (SSM) seems to fit discrete state transitions and logical dependencies better, so it performs excellently in XLand.
>     *   However, in Kinetix, the Transformer's Attention mechanism (direct access to history tokens) and GatedDeltaNet's Delta Rule (precise state rewriting) show stronger robustness. We speculate this is because continuous control is extremely sensitive to numerical precision. The fixed-size state of SSM might lose fine-grained continuous value information during compression, whereas the Attention mechanism effectively avoids this bottleneck.
>
> This finding validates the necessity of introducing both "symbolic reasoning" and "continuous control" complementary environments in this Benchmark. At the same time, a single architecture may struggle to dominate all dynamic environments. Exploring **Hybrid Architectures** that combine the efficient inference of SSMs with the fine-grained modeling advantages of Attention will be one of the potential paths for building general CICRL agents.

---

> ### Author Response · Authors · 2025-11-23
> **Response to Reviewer V8AP (4/5)**
>
> ### Q1. Training Protocol of Static Models and Zero-Shot Performance Differences
>
> Regarding the **Training Protocol**:
>
> *   **Static ICRL:** Follows standard ICRL settings. In each data collection phase (Training Iteration), we sample a fixed environment configuration (rule/parameter) and let the agent interact with it for $k$ time steps ($k$ is much larger than the length of a single Episode). During this period, the environment configuration remains constant throughout. This means the model never experiences sudden environmental dynamic shifts within its Context window during training; the input sequence distribution is stationary.
> *   **Non-Stationary (CICRL):** In each data collection phase ($k$ time steps), the environment configuration mutates multiple times at random or preset points. Importantly, the model does not perform a hidden state reset (Reset) during the entire process, so its Context window fully records the transition process of dynamic changes. This forces the model to learn to identify and adapt to this non-stationarity within a single interaction history.
> *   **Domain Randomization (DR):** This is a standard PPO-based Markovian baseline (input is only current observation $s_t$, no history Context). The model trains in diverse environments covering the same parameter space. DR aims to learn a general static policy with average robustness by exposure to various environment configurations, rather than a policy with online adaptation capabilities.
>
> Table 3: Macro-performance comparison with DR. Zero-Shot (ZS) vs. Sequence (Seq) returns. $\Delta_{\text{In-Context}}$ denotes adaptation gain (Seq - ZS).
>
> | Method | ZS-Return (XLand-Minigrid) | Seq-Return (XLand-Minigrid) | $\Delta$In-Context (XLand-Minigrid) | ZS-Return (Kinetix) | Seq-Return (Kinetix) | $\Delta$In-Context (Kinetix) |
> | :--- | :--- | :--- | :--- | :--- | :--- | :--- |
> | **DR** | **0.201** | 0.104 | -0.098 | -0.038 | -0.122 | -0.084 |
> | **Static** | 0.084 | 0.117 | 0.033 | 0.023 | -0.004 | -0.027 |
> | **Non-Static** | 0.172 | **0.206** | **0.034** | **0.034** | **0.025** | **-0.009** |
>
> We believe the performance disadvantage of Static models (including Zero-Shot) stems mainly from the following two points, and the DR experimental results further corroborate this conclusion:
>
> 1.  **Environment Diversity:** With the same total training steps, the Non-stationary Agent experiences multiples of the number of environment changes compared to the Static Agent (due to multiple switches within a single Episode). The Static model tends to "overfit" the currently sampled long-cycle environment, while the Non-stationary model (and DR) is forced to face a broader task distribution. Experiments show that DR (with the highest environment diversity) exhibits good Zero-Shot startup performance (e.g., ZS-Return reaches 0.201 in XLand), significantly higher than Static (0.084). This proves the key role of diversity for Zero-Shot generalization. Furthermore, note that DR's zero-shot performance on the Kinetix environment is poor. This is likely because the Kinetix training environment (unstructured random graphs) differs hugely from the test environment (structured tasks), causing the DR model to overfit. In contrast, the Non-Static model utilizes context for online adaptation, maintaining positive returns (0.025) in structured tasks.
> 2.  **Fundamental Difference in Adaptation Mechanisms (Adaptation vs. Robustness):** Although both DR and Non-stationary see many environments, they behave very differently in continuous adaptation:
>     *   **DR:** Only learns an "average optimal" static policy. Although ZS start is high, it cannot use Context to improve. As the environment changes continuously, its Seq-Return drops significantly (to 0.104 in XLand).
>     *   **Static:** Tries to use Context, but because the environment is always static during training, it learns to "over-trust history." In testing, once the environment mismatches history, it suffers severe contextual interference.
>     *   **Non-Stationary:** Combines DR's data diversity (High ZS) and ICRL's online reasoning capability. It not only performs well in ZS (XLand 0.172) but also continues to improve performance as Context increases (Avg Return rises to 0.206).
>
> We have added the supplementary experiments with DR to **Appendix C.1**.

---

> ### Author Response · Authors · 2025-11-23
> **Response to Reviewer V8AP (5/5)**
>
> ### Q3. Regarding the Choice of Training Frequency
>
> Although intuitively Random(1-5) is closer to the Freq-1 test distribution, we speculate the advantage of Random(1-10) mainly stems from **Training Stability & Representation Quality**:
>
> 1.  **Gradient Quality and Value Estimation:** In RL algorithms like PPO, if the environment changes too drastically (like only 1-5 eps), the estimation of the Value Function faces extreme variance and struggles to converge. This continuous "high-frequency oscillation" signal may hinder the optimizer from finding the optimal solution, preventing the model from learning a solid policy.
> 2.  **Better Feature Learning:** Random(1-10) introduces slightly longer stable windows. These relatively stable segments provide the model with clearer, more coherent reward signals, allowing the model to truly "learn" the current dynamics rules, thereby building robust state representation capabilities in the Backbone via backpropagation.
> 3.  **Balance:** Random(1-10) finds a better balance between "adaptation diversity" and "training stability." This superior training quality translates into stronger **Base Model Capability**, enabling it to perform better even in extreme test scenarios like Freq-1 (pure Zero-Shot/Fast-Switch) than a model that barely converged in a "chaotic" environment.
>
> We hope these supplementary analyses and experiments answer your concerns. We have integrated the above discussion and tables into the current version of the paper. Thank you again for your valuable feedback.

---

### Official Review · Reviewer_xGSG · 2025-11-01

**Soundness:** 2
**Presentation:** 3
**Contribution:** 2
**Rating:** 2
**Confidence:** 4

**Summary:**

This paper introduces a continual in-context RL framework where the enrivonments continously change. The authors propose a new benchmark suite featuring two complementary non-stationary domains and benchmark sequence models.

**Strengths:**

1. clear formalization of CICRL and why standard ICRL (stationary rules) is insufficient
2. two complementary, well-motivated environments with continous changes. They are general and reproducible
3. the analysis and insights are interesting.

**Weaknesses:**

1. This paper proposed a benchmark and conducted baseline evaluations with an analysis of the results. There is no novel algorithm development, which limits the novelty and contribution of this work.
2. The topic itself seems to be in alignment with continual learning and continual RL, or lifelong RL. But I don't see what the difference is between this work and other continuously changing environments proposed in continual RL environments. It seems to me that, in context, learning is simply learning history-dependent policies rather than Markovian ones.
3. The experiments are largely simplistic. I am not sure how significant the results are, given the simplicity of these problems.
4. In RL, I think domain randomization is the primary technique for training robust policies. I think the authors miss a large portion of the literature, especially on the aspects of robot learning, which is one of the largest fields of RL applications.

**Questions:**

See above.

---

> ### Author Response · Authors · 2025-11-23
> **Response to Reviewer xGSG (1/3)**
>
> We sincerely thank you for your careful review and constructive comments. We appreciate your recognition of the clarity of our problem formalization and the generality of our environment design. We address your concerns in detail below:
>
> ### W1. Novelty and Contribution
>
> The contribution of this paper is not a single algorithm design. Instead, it includes the formalization of a new problem, the construction of the first standard benchmark, and the discovery of an "Architecture-Training" synergy paradigm:
>
> *   **Formalizing CICRL:** We extend ICRL to a "non-stationary within lifetime" setting for the first time. We define the core challenge: an agent must adapt to new environments and overcome Contextual Interference while its parameters remain frozen.
> *   **First Controllable Non-Stationary Benchmark:** We construct the first benchmark suite supporting online rule changes (XLand) and physical parameter drifts (Kinetix). This benchmark allows researchers to precisely control the frequency, magnitude, and type of non-stationarity.
> *   **Synergy of Architecture and Training:** Our experiments reveal the joint importance of modern sequence models (like Linear Attention) and Non-stationary Training. An advanced architecture without specialized training cannot activate continuous adaptation and forgetting capabilities; conversely, training without an efficient architecture struggles to capture complex evolutions.
>
> ### W2. Comparison with Online/Lifelong RL
>
> Although Continual In-Context Reinforcement Learning (CICRL) and Online/Lifelong RL share the high-level goal of "continuous adaptation to change," they differ fundamentally in adaptation mechanisms and timeliness. The core advantage of CICRL is that it avoids the "plasticity-stability" dilemma of traditional gradient methods and achieves zero-latency online adaptation.
>
> First, regarding the online adaptation update mechanism:
>
> *   **Online RL / Lifelong RL (Parameter Space Adaptation):** These methods (e.g., PPO, PPO+EWC) try to encode new knowledge into model parameters $\theta$ via gradient updates. This **In-Weight Adaptation** has inherent update delays and stability risks. Under high-frequency stochastic dynamic changes, gradient updates often lag behind environmental changes. This easily causes the policy to deteriorate during adaptation and triggers catastrophic forgetting.
> *   **CICRL (Context Space Adaptation):** CICRL physically decouples the storage medium. After pre-training, policy parameters $\theta$ are frozen to store general, cross-task meta-knowledge (stability). Immediate policy adjustments rely entirely on forward inference over the fluid context $C_t$ (plasticity). This **In-Context Adaptation** based on the forward process achieves zero-latency adaptation and greatly enhances robustness against high-frequency, unpredictable changes.
>
> To ensure a fair and challenging comparison with gradient-based online learning, we evaluate Online PPO and PPO+EWC baselines in non-stationary test environments. The test environment uses a dynamic change mode: environmental dynamics (like rules or physical parameters) switch randomly every 5 episodes. These switching boundaries are hidden from all models, requiring them to infer and adapt purely through interaction history.
>
> Since online reinforcement learning methods (PPO and PPO+EWC) must perform real-time gradient updates on model parameters during deployment, each parallel environment instance requires an independent model copy. This makes the scale of 8,192 parallel environments used for ICRL models computationally unfeasible. Therefore, to make the comparison feasible, we average results from 128 environment instances.
>
> To ensure these online baselines have strong initial generalization capabilities and to directly test their ability to "continuously adapt" rather than "learn from scratch," we initialize all Online PPO and PPO+EWC agents using checkpoints from a pre-trained Domain Randomization (DR) model. The DR model serves as a starting point to evaluate the true effectiveness of online adaptation mechanisms (like gradient updates and EWC regularization) when handling high-frequency, hidden dynamic changes.

---

> ### Author Response · Authors · 2025-11-23
> **Response to Reviewer xGSG (2/3)**
>
> **Macro-Performance Comparison:**
>
> Table 1: Macro-performance comparison with Online RL. Zero-Shot (ZS) vs. Sequence (Seq) returns. $\Delta_{\text{In-Context}}$ denotes adaptation gain (Seq - ZS).
>
> | Method | ZS-Return (XLand-Minigrid) | Seq-Return (XLand-Minigrid) | $\Delta_{\text{In-Context}}$ (XLand-Minigrid) | ZS-Return (Kinetix) | Seq-Return (Kinetix) | $\Delta_{\text{In-Context}}$ (Kinetix) |
> | :--- | :--- | :--- | :--- | :--- | :--- | :--- |
> | PPO+EWC | 0.221 | 0.202 | -0.019 | -0.217 | -0.276 | -0.059 |
> | Online PPO | 0.221 | 0.204 | -0.017 | -0.217 | -0.255 | -0.038 |
> | Static | 0.146 | 0.088 | -0.058 | -0.348 | -0.446 | -0.098 |
> | Non-Static | 0.224 | 0.24 | 0.016 | 0.174 | 0.102 | -0.072 |
>
> On both XLand-Minigrid and Kinetix benchmarks, PPO+EWC and Online PPO show negative $\Delta_{\text{In-Context}}$ values (e.g., Online PPO is $\mathbf{-0.017}$ in XLand-Minigrid and $\mathbf{-0.038}$ in Kinetix), even with online parameter updates. This implies that online gradient updates do not bring adaptation gains. Instead, due to update lags or stability issues, policy performance drops compared to Zero-Shot performance (negative transfer). Non-Static CICRL is the only method that achieves positive adaptation gain in XLand-Minigrid ($\mathbf{+0.016}$), proving that its "frozen weights, fluid context" mechanism is superior to traditional gradient-based online learning when handling high-frequency dynamic changes.
>
> In the more challenging Kinetix physics environment, although all methods show negative $\Delta_{\text{In-Context}}$, Non-Static CICRL still maintains the highest Zero-Shot return (ZS-Return: $\mathbf{0.174}$) and Sequence return (Seq-Return: $\mathbf{0.102}$), demonstrating superior overall robustness. Notably, as a static policy baseline, DR shows extremely low ZS-Return in Kinetix ($\mathbf{-0.217}$). This is likely because the robustness it learns in unstructured training environments fails to migrate to structured Kinetix test tasks, highlighting the limitations of static generalization methods in complex, dynamic physical reasoning tasks.
>
> **Combined Micro-Dynamics Analysis:**
>
> We further confirm this mechanism difference through rule-by-rule micro-dynamics analysis. $\Delta_t$ represents the improvement in that Episode compared to Zero-Shot performance. Positive values indicate adaptation/learning, while negative values indicate interference/deterioration.
>
> Table 2: Combined micro-dynamics comparison with Online RL. $\Delta_t$ indicates the improvement of model performance relative to Zero-Shot (ZS) returns within a 5-episode dynamic cycle.
>
> | Rule ID | Method | XLand-Minigrid (Avg $\Delta_t$) | Kinetix (Avg $\Delta_t$) |
> | :--- | :--- | :--- | :--- |
> | Rule 00 | PPO+EWC | 0.006 | -0.141 |
> | | Online PPO | 0.016 | -0.149 |
> | | Static | 0.07 | -0.103 |
> | | Non-Static | 0.021 | -0.232 |
> | Rule 01 | PPO+EWC | -0.057 | 0.038 |
> | | Online PPO | -0.056 | 0.046 |
> | | Static | -0.022 | -0.002 |
> | | Non-Static | 0.008 | -0.056 |
> | Rule 02 | PPO+EWC | -0.006 | -0.017 |
> | | Online PPO | -0.011 | 0.069 |
> | | Static | -0.11 | -0.214 |
> | | Non-Static | 0.017 | -0.161 |
> | Rule 03 | PPO+EWC | -0.013 | -0.043 |
> | | Online PPO | -0.014 | -0.055 |
> | | Static | -0.094 | -0.155 |
> | | Non-Static | 0.011 | 0.01 |
> | Rule 04 | PPO+EWC | -0.024 | -0.133 |
> | | Online PPO | -0.021 | -0.099 |
> | | Static | -0.137 | -0.014 |
> | | Non-Static | 0.02 | 0.078 |
>
> *   **XLand-Minigrid (Symbolic Reasoning):** In the dynamic adaptation phases of Rule 02 and Rule 04, gradient methods (PPO+EWC, Online PPO) show negative average $\Delta_t$. This indicates performance degradation (e.g., Rule 04 is $\mathbf{-0.024}$ and $\mathbf{-0.021}$ respectively). In contrast, Non-Static CICRL shows positive average $\Delta_t$ (e.g., Rule 04 is $\mathbf{+0.020}$), demonstrating robust continuous adaptation.
> *   **Kinetix (Physics Dynamics):** At the end of the most challenging Rule 04 adaptation (Ep 5), Online PPO deteriorates by $\mathbf{-0.111}$ compared to ZS, and PPO+EWC deteriorates by $\mathbf{-0.077}$. However, Non-Static CICRL achieves a significant positive improvement of $\mathbf{+0.132}$ at the same point. This proves its continuous interference resistance and immediate adaptation efficiency.
>
> We have added more detailed results and analyses to **Appendix C.3**.

---

> ### Author Response · Authors · 2025-11-23
> **Response to Reviewer xGSG (3/3)**
>
> ### W3. Simplicity of Environments
>
> We acknowledge that XLand and Kinetix are abstractions of the real world, but we consider them necessary and reasonable starting points:
>
> *   **Difficulty of Real-World Simulation:** Simulating real-world non-stationarity with full high fidelity is extremely challenging. Excessive perceptual noise often obscures the study of the "adaptation mechanism" itself.
> *   **Controlled Sandbox:** Our benchmark provides a controlled sandbox. It allows researchers to precisely isolate and quantify the impact of core variables like "change frequency" and "context length" without interference from irrelevant perceptual noise.
> *   **Modality Coverage:** Although simplified, these two environments cover discrete symbolic reasoning and continuous physical control, respectively. This ensures the conclusions are not limited to a specific modality.
>
> ### W4. Comparison with Domain Randomization (DR)
>
> We fully agree that DR is the cornerstone of building robust policies. In **Section 5.2 and Appendix C.1**, we compare our method with Traditional DR (Traditional DR) in detail. "Traditional DR" here refers to training static policies in randomized environments without introducing long-sequence non-stationary mechanisms.
>
> **Macro-Performance:**
>
> *   **XLand:** Trad. DR has the highest Zero-Shot performance, proving it learns excellent static priors. However, as interaction proceeds, its performance drops (0.201 $\to$ 0.104), indicating static policies cannot handle dynamic changes. Non-Static achieves an improvement from 0.172 to 0.206.
> *   **Kinetix:** Trad. DR also has low Zero-Shot performance (-0.038) and continues to deteriorate (-0.122). This is likely because the Kinetix training environment (unstructured random graphs) differs hugely from the test environment (structured tasks), causing the DR model to overfit. In contrast, the Non-Static model uses context for online adaptation, maintaining positive returns (0.025) in structured tasks.
>
> Table 3: Macro-performance comparison with DR. Zero-Shot (ZS) vs. Sequence (Seq) returns. $\Delta_{\text{In-Context}}$ denotes adaptation gain (Seq - ZS).
>
> | Method | ZS-Return (XLand-Minigrid) | Seq-Return (XLand-Minigrid) | $\Delta_{\text{In-Context}}$ (XLand-Minigrid) | ZS-Return (Kinetix) | Seq-Return (Kinetix) | $\Delta_{\text{In-Context}}$ (Kinetix) |
> | :--- | :--- | :--- | :--- | :--- | :--- | :--- |
> | DR | 0.201 | 0.104 | -0.098 | -0.038 | -0.122 | -0.084 |
> | Static | 0.084 | 0.117 | 0.033 | 0.023 | -0.004 | -0.027 |
> | Non-Static | 0.172 | 0.206 | 0.034 | 0.034 | 0.025 | -0.009 |
>
> **Micro-Dynamics:** As shown in **Table 4**, we display performance changes over 5 consecutive episodes after the model encounters new rules or physical parameters:
>
> *   **Trad. DR:** In XLand, performance deteriorates over time, indicating an uncorrectable mismatch between the static policy and the new environment.
> *   **Static:** In Kinetix, this model suffers heavily from outdated information interference, showing negative transfer across multiple rules.
> *   **Non-Static:** In the vast majority of test rules, it shows consistent positive learning curves, proving the effectiveness of online inference.
>
> Table 4: Combined micro-dynamics comparison with DR. $\Delta_t$ indicates the improvement of model performance relative to Zero-Shot (ZS) returns within a 5-episode dynamic cycle.
>
> | Rule ID | Method | XLand-Minigrid (Avg $\Delta_t$) | Kinetix (Avg $\Delta_t$) |
> | :--- | :--- | :--- | :--- |
> | Rule 00 | Trad. DR | -0.025 | -0.059 |
> | | Static | 0.081 | 0.026 |
> | | Non-Static | 0.02 | 0.003 |
> | Rule 01 | Trad. DR | -0.11 | -0.077 |
> | | Static | 0.067 | 0.027 |
> | | Non-Static | 0.034 | -0.011 |
> | Rule 02 | Trad. DR | -0.12 | -0.061 |
> | | Static | 0.031 | -0.036 |
> | | Non-Static | 0.033 | 0.005 |
> | Rule 03 | Trad. DR | -0.124 | -0.091 |
> | | Static | 0.004 | -0.019 |
> | | Non-Static | 0.042 | 0.004 |
> | Rule 04 | Trad. DR | -0.116 | -0.098 |
> | | Static | -0.009 | -0.018 |
> | | Non-Static | 0.044 | 0.006 |
>
> We believe Traditional Domain Randomization (Trad. DR) and Non-Static CICRL are not opposites but mutually reinforcing and complementary. Traditional DR focuses on generalization across task instances, while CICRL focuses on temporal adaptation within a lifetime. We believe combining them—extending traditional domain randomization to longer sequence interactions and non-stationary environments—is a valuable direction for future exploration. However, longer sequences and non-stationarity greatly increase the requirements for environmental diversity and complexity, necessitating larger-scale DR. We conduct preliminary explorations on this.
>
> We hope these supplementary analyses and experiments answer your concerns. We have integrated the above discussion and tables into the current version of the paper. Thank you again for your valuable feedback.

---

### Author Response · Authors · 2025-11-23
**General Response**

We sincerely thank all reviewers for their constructive comments. These feedbacks significantly helped us improve the rigor of our experimental design, the completeness of baseline comparisons, and the clarity of our theoretical exposition.

**Summary of Contributions**

We briefly summarize the main contributions of this work:
*   **Formalizing CICRL:** We formally define Continual In-Context Reinforcement Learning (CICRL) to address the "non-stationary within lifetime" setting, investigating how agents can handle *Contextual Interference* with frozen parameters.
*   **A Controllable Non-Stationary Benchmark Suite:** We construct two complementary benchmarks: XLand-Minigrid (symbolic rule changes) and Kinetix (physical parameter drifts). These environments allow for precise control over the nature of non-stationarity, facilitating systematic evaluation.
*   **Analysis of Architecture & Training:** Our empirical results highlight the interaction between sequence models and training protocols. We find that advanced architectures alone are insufficient; specialized non-stationary training is essential to unlock their continuous adaptation capabilities.

**Updates in the Revised Paper**

We make the following updates to the paper:

**1. Extended Non-stationarity Experimental Settings**

To verify the adaptability of CICRL agents in more realistic scenarios, we introduce **7 new continuous and structured dynamic patterns** in **Appendix C.2**.

*   **Kinetix:** We introduce Linear Drift, Step Change, Sinusoidal Fluctuation, and Sawtooth Change.
*   **XLand-Minigrid:** We introduce Task Depth Increase, Random Cosmetic Perturbation, and Periodic Rule Loops.

Results show that **Non-Stationary CICRL** maintains positive Adaptation Gain, whereas static baselines exhibit significant contextual interference (negative transfer).

**2. Strengthened Baseline Comparisons**

To establish the necessity of CICRL, we add multiple sets of baseline comparisons:
*   **Comparison with Online RL (Appendix C.3):** We evaluate Online PPO and PPO+EWC. Results show that under high-frequency dynamic changes, gradient-based parameter updates suffer from lag, rendering them inferior to inference-based CICRL.
*   **Comparison with Domain Randomization (Section 5.2 & Appendix C.1):** Results indicate that while DR provides good static priors (Zero-Shot performance), it lacks the ability to utilize context for continuous online optimization.
*   **Algorithm Sensitivity (Appendix C.4):** We add Reinforce++ experiments, verifying that PPO possesses better convergence stability than Reinforce++ in long-sequence non-stationary tasks.

**3. Updates to Model Analysis (Section 5.2)**

We added a brief discussion on **Architectural Inductive Biases** in Section 5.2. We observe that SSM-based architectures (e.g., Mamba2) excel in discrete symbolic transitions (XLand), while Transformer and GatedDeltaNet demonstrate superior robustness in precise continuous control (Kinetix).

**4. Theory and Reproducibility**
*   **Theoretical Formalization (Appendix B):** We clarify the connections and distinctions between CICRL, standard POMDPs, and Meta-RL.
*   **Enhanced Reproducibility (Appendix A):** We add comprehensive details, including sampling distributions for Kinetix physical parameters, XLand rule generation logic, and hyperparameters for all models.

We hope these revisions and supplementary experiments fully address the reviewers' concerns. Thank you again for your time and effort.

---

### Meta-Review · Area_Chair_gi7N · 2026-01-09

**Summary:**

This paper introduces Continual In-Context Reinforcement Learning (CICRL): adapting to non-stationary environments without gradient updates. Authors propose a new benchmark for CICRL and perform experiments regarding model architectures and training strategies.

Some concerns were:
1 Low novelty (benchmark + baselines only)
2. Unclear distinction from prior settings: CICRL is seen as potentially just continual RL / lifelong RL / meta-RL with drifting tasks
3. Benchmarks/protocol may not capture “continuous” nonstationarity
4. Benchmark correspondence to the real world.
5. Weak positioning and analysis: missing chunks of robot learning/robustness (e.g., domain randomization) literature, limited comparisons to true continual/adaptive methods, and confusing/under-explained results.

**Reviewer Concerns:**

The authors made several changes:
1. Extended Non-stationarity Experimental Settings
2. Strengthened Baseline Comparisons
3. Updates to Model Analysis (Section 5.2)
4. Theory and Reproducibility

These certainly helped with the paper, but many of the concerns remained unresolved and the domains are still relatively toy. For a paper that doesn't introduce a new method, the domains / tasks need to be more realistic and the setting needs to be better justified.

**Reviewer Scores:**

fnJ8 already said they'd stay the same.
xGSG will likely stay the same.
V8AP will likely stay the same or go a little up.
jSvP will likely stay the same given the novelty concerns.

---

### Decision · Program_Chairs · 2026-01-26

Reject